# HGM³: Hierarchical Generative Masked Motion Modeling with Hard Token Mining

**Minjae Jeong**[*]    **Yechan Hwang**[*]    **Jaejin Lee**    **Sungyoon Jung**    **Won Hwa Kim**
Pohang University of Science and Technology (POSTECH), Pohang, South Korea
{minjaetidtid, yechan99, jlee4923, syjung, wonhwa}@postech.ac.kr

## ABSTRACT

Text-to-motion generation has significant potential in a wide range of applications including animation, robotics, and AR/VR. While recent works on masked motion models are promising, the task remains challenging due to the inherent ambiguity in text and the complexity of human motion dynamics. To overcome the issues, we propose a novel text-to-motion generation framework that integrates two key components: Hard Token Mining (HTM) and a Hierarchical Generative Masked Motion Model (HGM³). Our HTM identifies and masks challenging regions in motion sequences and directs the model to focus on hard-to-learn components for efficacy. Concurrently, the hierarchical model uses a semantic graph to represent sentences at different granularity, allowing the model to learn contextually feasible motions. By leveraging a shared-weight masked motion model, it reconstructs the same sequence under different conditioning levels and facilitates comprehensive learning of complex motion patterns. During inference, the model progressively generates motions by incrementally building up coarse-to-fine details. Extensive experiments on benchmark datasets, including HumanML3D and KIT-ML, demonstrate that our method outperforms existing methods in both qualitative and quantitative measures for generating context-aware motions.

## 1 INTRODUCTION

Text-to-motion generation, which creates 3D human motion from textual descriptions, has recently emerged as an important task for a wide range of applications such as animation, graphics, robotics, and AR/VR (Azadi et al., 2023; Guo et al., 2024). As creating dynamic scenes with various motions from scratch is costly and time-consuming, text-to-motion generation offers an efficient solution by integrating natural language interfaces into the motion creation. Despite its potential, the task is still challenging because of the intrinsic ambiguity of human language and the complicated structure of human motion dynamics. To tackle these challenges, various deep generative models have been proposed, including methods based on language-motion latent space alignment (Ahuja & Morency, 2019), diffusion models (Tevet et al., 2023), and autoregressive models (Zhang et al., 2023).

Recently, generative masked models have become particularly popular for human motion generation (Pinyoanuntapong et al., 2024b; Guo et al., 2024), as they offer a powerful approach for generating realistic motion sequences from tokenized motion representations. These models first convert continuous motion data into discrete motion tokens, where certain tokens in the sequence are randomly masked. The models are then trained to infer the missing tokens based on the unmasked ones, enhancing their understanding of spatial and temporal relationships of action-specifics within the motion and ensuring the smooth continuity of the generated sequences. Such frameworks have proven effective in generating human motion, even from incomplete or noisy input data.

These masked motion generation models, however, rely on random masking strategies, which do not effectively target the most informative components in the data. Such approaches do not consider the varying levels of difficulty within motion sequences, leading to suboptimal learning where both simple and challenging parts are treated evenly. Indiscriminate masking may result in the model being insufficiently challenged during training, limiting its capacity to develop a deep understanding of difficult motion patterns. Despite the success of advanced masking techniques in other domains

---

[*]These authors contributed equally to this work.

such as masking hard patches in images (Wang et al., 2023a), there has been limited exploration of how these techniques could be adapted to improve the learning process in human motion generation.

Moreover, previous text-to-motion generative models typically rely on extracting a single sentence embedding from language processing methods such as CLIP (Radford et al., 2021) to represent motion-from-text, which can be potentially problematic for generating complex and sequential motions. While these embeddings are suitable for static data such as images, human motion inherently involves a dynamic sequence of actions and interactions, which cannot be fully captured by a single vector representation. This results in a lack of detailed control and expressiveness in the generated motions, particularly for complex or nuanced sequences.

To overcome the limitations listed above, we propose 1) a Hard "Token" Mining (HTM) strategy and 2) integrating it with hierarchical semantic graph-based textual conditioning. Inspired by image patch mining (Wang et al., 2023a), our HTM aims to selectively mask the most challenging tokens within a motion sequence. Implemented as a teacher-student scheme, the teacher model guides the student model to learn from the most challenging motion tokens by selective masking. HTM pushes the limits beyond random masking methods by focusing on challenging regions, thereby significantly improving the model's ability to capture complex temporal (i.e., the progression and timing of actions) and spatial (i.e., how different joints move in relation to each other) dependencies. Additionally, we introduce a **H**ierarchical **G**enerative **M**asked **M**otion **M**odel (HGM$^3$), which employs a hierarchical semantic graph from Shi & Lin (2019) for a masked motion model to represent input text at various semantic granularities. Originally developed for natural language understanding, this representation allows our model to incorporate contextual information into the motion generation, improving the details and relevance within generated motion sequences.

To this end, implementation of our key ideas as an integrated framework leads to the following **contributions**: **1)** We propose Hard Token Mining (HTM) strategy, which selectively masks challenging regions in motion sequences to enhance generation performance over traditional random masking approaches, **2)** We introduce a hierarchical semantic graph representation into generative masked motion model that organizes text semantics into various granularity, which allows the model to incorporate multiple layers of context and lead to more accurate and context-aware motion generation. **3)** We showcase the effectiveness of our model through both qualitative and quantitative evaluations, which demonstrate state-of-the-art results in standard text-to-motion generation tasks.

## 2 RELATED WORK

### 2.1 TEXT-DRIVEN MOTION GENERATION

Early methods for text-driven human motion generation focus on aligning the latent representations of motion and text by minimizing the Kullback-Leibler (KL) divergence or contrastive loss between their distributions (Ahuja & Morency, 2019; Tevet et al., 2022; Petrovich et al., 2022; Ghosh et al., 2021; Guo et al., 2022c). However, aforementioned methods suffer from unnatural motion generations due to the implicit discrepancy between the representation of the motion and text. To address this, learning the stochastic mapping between motions and texts was proposed. T2M (Guo et al., 2022b) utilized a temporal VAE to learn the mapping, and diffusion models such as MDM (Tevet et al., 2023) trained a transformer (Vaswani, 2017) encoder to reconstruct noised raw motion sequences, while MLD (Chen et al., 2023) leveraged latent motion representations to enhance the computational efficiency of MDM. GraphMotion (Jin et al., 2024) introduced hierarchical text conditionings defined in three semantic levels to gain finer control over motion generation. Other works (Kong et al., 2023; Wang et al., 2023b; Dabral et al., 2023; Zhang et al., 2024; Huang et al., 2024; Dai et al., 2025) also used the diffusion model for text-driven motion generation.

Meanwhile, motion generation with masked transformers have shown better efficiency. T2M-GPT (Zhang et al., 2023) and MotionGPT (Jiang et al., 2023) generated motion tokens autoregressively from masked tokens, while MMM (Pinyoanuntapong et al., 2024b) employed random masking on input motion tokens. MoMask (Guo et al., 2024) used a residual VQ-VAE to reconstruct the final motion sequence using a residual transformer. BAMM (Pinyoanuntapong et al., 2024a) adopted a bidirectional causal masking to complement autoregressive transformers.

### 2.2 MASKING STRATEGIES IN MASKED MODELING

Masking strategies have been widely explored for masked modeling to obtain more generalizable representation. BERT (Devlin et al., 2019) introduces a random masking strategy on the input text tokens, where bidirectional transformers predict the masked tokens. ViT (Dosovitskiy et al., 2021)

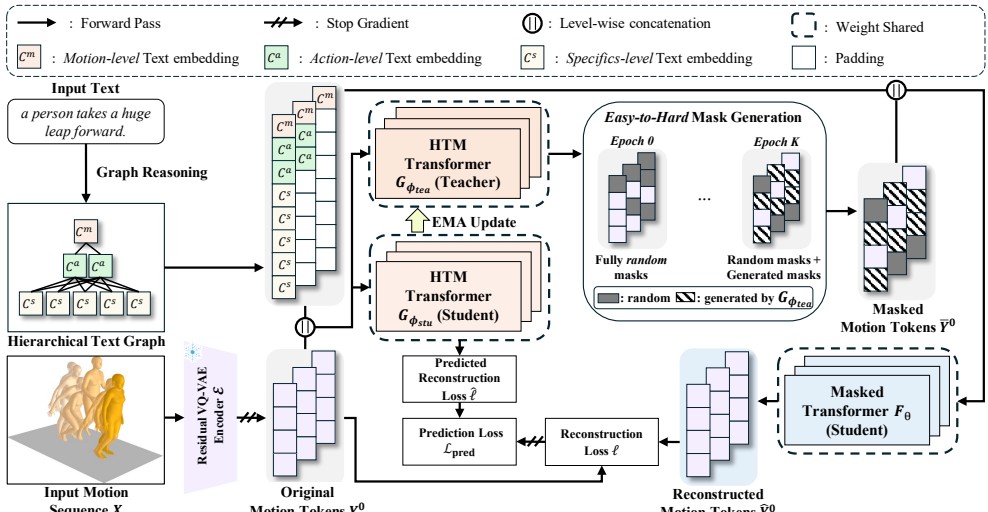

Figure 1: **Overview of Training HGM³.** The input motion $X$ is encoded into latent features using a pre-trained residual VQ-VAE Encoder $\mathcal{E}$, which are then quantized into discrete motion tokens $Y^0$, while the input text is processed through CLIP and GAT to generate a hierarchical graph embedding with motion-, action-, and specifics-level nodes as conditioning inputs. The framework uses a teacher-student paradigm, where the teacher $G_{\phi_{tea}}$ is updated by the student $G_{\phi_{stu}}$ and identifies challenging tokens to guide the masked transformer $F_\theta$. The $F_\theta$ reconstructs the motion tokens $\hat{Y}^0$, from which the token-wise reconstruction loss $\ell$ is computed. The loss $\ell$, along with the student-predicted loss $\hat{\ell}$, is then used to obtain the prediction loss $\mathcal{L}_{\text{pred}}$.

extends similar random masking strategy to image patches. BEiT (Bao et al., 2022) separately masks image patches and tokenizer tokens, using blockwise masking for the patches and random masking for the tokens. MAE (He et al., 2022) showed that masking a large portion of the entire image patches helps reduce the spatial redundancy between them, and SimMIM (Xie et al., 2022) used larger masks to learn good representations regardless of the masking ratio. Other works adopting random masking include MaskGIT (Chang et al., 2022) and Muse (Chang et al., 2023).

Rather than applying masks at totally random, some approaches leverage the semantic patterns of the input or learned strategies. SemMAE (Li et al., 2022) uses a part segmentation map to guide the masking process, which is derived by selecting the most prominent features from the self-attention maps. ADIOS (Shi et al., 2022) learns how to mask using a U-Net model with an adversarial training objective. Hard Patch Mining (HPM) (Wang et al., 2023a) employs a teacher-student model that transitions from completely random masking to focusing on difficult image patches identified by high reconstruction loss predictions from the teacher model. Inspired by HPM, we propose an effective masking strategy, i.e., HTM, that can be applied to motion tokens.

## 3 METHOD

Our goal is to develop a comprehensive text-to-motion synthesis framework that generates high-quality and contextually accurate human motion based on textual descriptions. To realize this, our approach integrates multiple components that jointly enhance both the representation and generation of motion data, as illustrated in Fig. 1. First, we utilize a motion tokenizer based on a residual VQ-VAE (Van Den Oord et al., 2017; Guo et al., 2024), which converts raw 3D human motion data into a sequence of discrete tokens through a hierarchical quantization process (Sec. 3.1). Next, we implement a masked transformer with a HTM strategy, which learns to mask discriminative parts of the motion and helps the model better recognize significant features in the data (Sec. 3.2). Furthermore, our framework introduces a hierarchical semantic graph-based textual conditioning for masked transformer, which structures semantic information into multiple levels to enhance the model's comprehension of textual descriptions (Sec. 3.3). Finally, we use a hierarchical inference process for motion generation, where the model iteratively refines the generated motion and employs a pre-trained residual transformer to correct quantization errors (Sec. 3.4).

### 3.1 MOTION TOKENIZER: RESIDUAL VQ-VAE

A VQ-VAE is conventionally used to encode motion sequences into a discrete latent space, which produces motion tokens that serve as inputs for a Transformer model (Vaswani, 2017). VQ-VAE

quantizes the continuous output of the encoder into a discrete latent space through a single vector quantization layer, which leads to information loss. To better approximate the encoder output, we used a residual VQ-VAE to quantize the lost information across additional $V$ quantization layers. We train a separate single residual transformer following Guo et al. (2024) and Pinyoanuntapong et al. (2024a) to predict the token sequences at the additional layers. The details are as follows.

Given a motion sequence $X = x_{1:N} \in \mathbb{R}^{N \times D}$ of length $N$ with a frame dimension of $D$, let $\mathcal{E}(X) = Z^0 = z^0_{1:n} \in \mathbb{R}^{n \times d}$ denote the output latent feature of the VAE encoder $\mathcal{E}$, where $N/n$ is the downsampling rate and $d$ is the latent dimension. The input and output of the $v$-th quantizer, $Z^v, \hat{Z}^v \in \mathbb{R}^{n \times d}$, are defined as follows:

$$\hat{Z}^v = Q(Z^v), \quad Z^{v+1} = Z^v - \hat{Z}^v, \tag{1}$$

where $Q(\cdot)$ represents the quantization operation, which maps each vector $z^v_i$ of the continuous latent feature $Z^v$ to its nearest entry in the codebook of $v$-th quantization layer. The reconstructed motion sequence through VAE decoder $\mathcal{D}$ is then given by $\hat{X} = \mathcal{D}\left(\sum_{v=0}^{V} \hat{Z}^v\right)$, and the training objective of the residual VQ-VAE is formulated as:

$$\mathcal{L}_{\text{rvq}} = \|X - \hat{X}\|_1 + \beta \sum_{v=0}^{V} \|Z^v - \text{sg}[\hat{Z}^v]\|_2^2, \tag{2}$$

where $\beta$ is a hyperparameter for the commitment loss, and $\text{sg}(\cdot)$ denotes the stop-gradient which blocks the gradient flow during backpropagation. The first term of $\mathcal{L}_{\text{rvq}}$ is optimized using a straight-through gradient estimator (Van Den Oord et al., 2017), and our codebooks are updated using exponential moving averages and codebook resets, as in Zhang et al. (2023) and Guo et al. (2024).

As a result, each motion sequence $X = x_{1:N}$ is quantized into $V + 1$ sequences $[\hat{z}^v_{1:n}]^V_{v=0}$ via the residual VQ-VAE, and the sequences are converted into indices $[y^v_{1:n}]^V_{v=0}$ with $y^v_{1:n} \in \mathbb{R}^n$, where each element of $y^v_{1:n}$ corresponds to an index in the $v$-th learned codebook. The first sequence contains the most information about the input motion and is used to train masked transformer, while the remaining sequences progressively contain less but still essential information and are used to train residual transformer. Detailed descriptions of the training and inference procedures for the residual VQ-VAE and the residual transformer are provided in Appendix A.

## 3.2 Masked Transformer with Hard Token Mining Strategy

Given a sequence of motion tokens $Y^0$ obtained from a pre-trained residual VQ-VAE, we apply masking to a subset of the tokens and train a masked transformer to reconstruct the original sequence. While existing methods such as Guo et al. (2024) and Pinyoanuntapong et al. (2024b) randomly select tokens to mask, we strategically choose the tokens to mask with criteria.

Our framework employs a teacher-student scheme, where the teacher model dynamically generates challenging masked token sequences to guide the student model. As shown in Fig. 1, the proposed model consists of a student model $(F_\theta, G_{\phi_{stu}})$ and a teacher model $(G_{\phi_{tea}})$. Here, $F_\theta(\cdot)$ represents the masked transformer parameterized by $\theta$, that reconstructs the masked tokens, while $G_\phi(\cdot)$ is an auxiliary transformer parameterized by $\phi$, which identifies challenging tokens for $F_\theta(\cdot)$ by predicting the relative ordering of reconstruction losses.

**Token Reconstructor.** Let $\bar{Y}^0 = \bar{y}^0_{1:n}$ be the masked motion token sequence. In this sequence, the tokens identified as challenging by the loss predictor $G_{\phi_{tea}}$ are replaced by a special `[Mask]` token, while all other tokens remain unchanged from $Y^0$. We denote the set of indices of these masked tokens as $\mathcal{M}$, which is dynamically updated at every epoch $t$. A masked transformer $F_\theta(\cdot)$ is trained on the masked sequence $\bar{Y}^0$ with contextual text embeddings $C$, which capture semantic information from the input text. Using both $\bar{Y}^0$ and $C$, the masked transformer predicts the corrupted parts of the motion tokens. The reconstruction loss for each masked token is defined as:

$$\ell_k = -\log F_\theta(y^0_k | \bar{Y}^0, C), \tag{3}$$

where $k \in \mathcal{M}$ denotes the index of each masked token. The total reconstruction loss is then computed as:

$$\mathcal{L}_{\text{rec}} = \sum_{k \in \mathcal{M}} \ell_k = \sum_{k \in \mathcal{M}} -\log F_\theta(y^0_k | \bar{Y}^0, C). \tag{4}$$

**Loss Predictor.** To identify motion tokens that are more difficult to reconstruct, a loss predictor is trained to predict $\texttt{argsort}(\ell)$ using a relative loss (Wang et al., 2023a). The model is trained to rank the reconstruction difficulty by predicting which token has the higher reconstruction loss for each pair of tokens $(i, j)$. Given $Y^0$ and $C$, $G_{\phi_{stu}}$ is optimized by computing the loss over only the tokens that are masked, i.e., those in $\mathcal{M}$. The prediction loss $\mathcal{L}_{\text{pred}}$ is defined as:

$$\mathcal{L}_{\text{pred}} = -\sum_{i \in \mathcal{M}} \sum_{\substack{j \in \mathcal{M} \\ j \neq i}} \mathbb{1}_{ij} \log\left(\sigma(\hat{\ell}_i - \hat{\ell}_j)\right) - \sum_{i \in \mathcal{M}} \sum_{\substack{j \in \mathcal{M} \\ j \neq i}} (1 - \mathbb{1}_{ij}) \log\left(1 - \sigma(\hat{\ell}_i - \hat{\ell}_j)\right), \tag{5}$$

where $\hat{\ell} = G_{\phi_{stu}}(Y^0, C)$ represents the predicted relative reconstruction loss for each token from the student and $\sigma(\cdot)$ is a sigmoid function. The indicator $\mathbb{1}_{ij}$ represents whether the value of $\ell_i$ is greater or smaller than $\ell_j$, i.e., $\mathbb{1}_{ij} = 1$ if $\ell_i > \ell_j$ and 0 otherwise. Here, $\mathcal{L}_{\text{rec}}$ is detached from the gradient, serving as the ground truth for loss prediction. To ensure stable predictions from the teacher model, a momentum update (He et al., 2020) is applied as:

$$\phi_{tea} \leftarrow \lambda \phi_{tea} + (1 - \lambda)\phi_{stu}, \tag{6}$$

where $\lambda$ represents the momentum coefficient. The student model is trained with two objectives: reconstruction loss and prediction loss. These two objectives are combined as:

$$\mathcal{L} = \mathcal{L}_{\text{rec}} + \mathcal{L}_{\text{pred}}, \tag{7}$$

where they alternate and complement each other. This interaction gradually encourages the student model to reconstruct challenging motion tokens, which leads to desired feature representation.

**Easy-to-Hard Mask Generation.** To help the model gradually adapt to more complex patterns and improve its ability to handle difficult tokens, we employ an Easy-to-Hard Mask Generation strategy. The masked regions are initially selected completely at random, gradually increasing the intensity by masking demanding tokens as training advances. This strategy ensures that the model starts by learning basic patterns and gradually adapts to more challenging parts of the data. In each iteration, the teacher model $G_{\phi_{tea}}$ predicts the relative reconstruction loss for each token, which is then ranked in descending order of difficulty using an $\texttt{argsort}(\cdot)$ operation. Tokens with the highest reconstruction loss are then selected for masking. For each epoch $t$, a proportion $\alpha_t$ of the tokens to be masked is selected based on this ranking, while the remaining $1 - \alpha_t$ tokens are chosen randomly. Throughout the training, the proportion of hard tokens increases linearly as:

$$\alpha_t = \alpha_0 + \frac{t}{T}(\alpha_T - \alpha_0), \tag{8}$$

where $\alpha_0$ and $\alpha_T$ are within the range $[0, 1]$, representing the initial and final proportion of hard tokens, respectively, and $T$ is the total number of training epochs. Consequently, at each epoch, $\alpha_t \cdot \gamma(\tau_t) \cdot n$ tokens with the highest predicted reconstruction loss are masked, while the remaining $(1 - \alpha_t) \cdot \gamma(\tau_t) \cdot n$ tokens are masked randomly. Here, we adopt a cosine function to schedule the masking ratio following Chang et al. (2022), defined as $\gamma(\tau_t) = \cos\left(\frac{\pi \tau_t}{2}\right) \in [0, 1]$, where $\tau_t$ is randomly sampled from a uniform distribution $\mathcal{U}(0, 1)$ during training.

## 3.3 Hierarchical Semantic Textual Conditioning for Masked Transformer

Existing works that generate motions from discrete tokens using transformers (Zhang et al., 2023; Guo et al., 2024; Pinyoanuntapong et al., 2024a) rely on a single implicit embedding as a condition token for a given text. This approach overlooks the fine-grained details of the text. To address this, we propose the Hierarchical Generative Masked Motion Model (HGM³), which leverages enhanced text embeddings as input conditions with HTM. Inspired by Chen et al. (2020) and Jin et al. (2024), we decompose the text into semantically fine-grained components to construct a hierarchical graph and obtain enhanced text embeddings through a Graph Attention Network (GAT) (Veličković et al., 2018). These embeddings are fed into the HGM³ facilitating a deeper understanding of the text.

We employ a semantic role parsing toolkit (Shi & Lin, 2019) to extract a hierarchical graph with three types of nodes, i.e., 1) motions, 2) actions, and 3) specifics, and twelve types of edges representing various relationships between nodes. The sentence is treated as a global motion node, with verbs as action nodes and attribute phrases as specific nodes. The motion node is connected to all action nodes, while each action node is linked to its corresponding specific nodes. For further details on the semantic role parsing process, please refer to Appendix B.

**Graph Reasoning.** To obtain text embeddings for each node, we use a pre-trained CLIP, and attention-based message passing is performed between neighboring nodes using GAT. From CLIP, we obtain node embeddings $c = [c^m, c^a, c^s]$, where $c^m \in \mathbb{R}^{d_c}$ represents the embedding of a single motion node, $c^a \in \mathbb{R}^{n_a \times d_c}$ denotes the embeddings of $n_a$ action-level nodes, and $c^s \in \mathbb{R}^{n_s \times d_c}$ corresponds to the embeddings of $n_s$ specific-level nodes, each with a dimension of $d_c$. Each node embedding is then transformed into $h = \{h^m, h^a, h^s\}$ through a shared weight $W \in \mathbb{R}^{d_c \times d_c}$ as:

$$h^m = c^m W, \quad h^a = c^a W, \quad h^s = c^s W. \tag{9}$$

For connected nodes $i$ and $j \in \mathcal{N}_i$, where $\mathcal{N}_i$ represents the set of neighbor nodes for node $i$, we concatenate the embeddings of the two nodes to obtain $\tilde{h}_{ij} = [h_i; h_j] \in \mathbb{R}^{2d_c}$. Let $M \in \mathbb{R}^{2d_c}$ be a shared transformation vector for all edge types, and $M_r \in \mathbb{R}^{2d_c \times E}$ be a relationship embedding matrix with distinct weights for $E$ edge types. A one-hot vector $r_{ij} \in \mathbb{R}^E$ represents the edge type between nodes $i$ and $j$. The attention coefficient $\tilde{e}_{ij}$ is then obtained as:

$$e_{ij} = \text{LeakyReLU}(M^\top \tilde{h}_{ij}) + \text{LeakyReLU}(r_{ij} M_r^\top \tilde{h}_{ij}), \quad \tilde{e}_{ij} = \frac{\exp(e_{ij})}{\sum_{k \in \mathcal{N}_i} \exp(e_{ik})}. \tag{10}$$

Finally, the output node embedding $C_i$ from GAT is computed as follows:

$$C_i = \psi \left( \sum_{j \in \mathcal{N}_i} \tilde{e}_{ij} h_j \right) + c_i, \tag{11}$$

where $\psi$ denotes a non-linear activation function.

**Textual Conditioning for Masked Transformer.** Eq. (4) and Eq. (5) are derived using $C^m$ as the only condition. Leveraging the three levels of text embeddings obtained from Eq. (11), we construct three types of conditions for each sentence: $C^m$, $[C^m; C^a]$, and $[C^m; C^a; C^s]$, where [;] denotes concatenation. The lengths of the condition tokens are aligned with padding tokens, and position encoding (Vaswani, 2017) is applied to the entire input sequence, including the condition tokens. These conditions are prepended to the motion token sequence and provided as input to the models, as illustrated in Fig. 1. Instead of using only $[C^m; C^a; C^s]$, providing all three types of conditions ensures that as the number of nodes increases from the root (motion level) to the leaves (specific level) of the semantic graph, the global meaning encapsulated by $C^m$ is not diminished, thereby maintaining a balance between global and fine-grained information.

These three types of conditions are used not only to train $F_\theta$, but also $G_\phi$. This is because we assume that the tokens that are difficult for $F_\theta$ to predict will change depending on the type of condition given. Taking this into account, the losses are redefined as follows. Let the token-wise reconstruction losses predicted by $G_{\phi_{stu}}$ with $C^m$, $[C^m; C^a]$, and $[C^m; C^a; C^s]$ be denoted as $\hat{\ell}^m$, $\hat{\ell}^a$, and $\hat{\ell}^s$, respectively. Using these losses, $\mathcal{L}_{\text{pred}}$ is computed separately as $\mathcal{L}_{\text{pred}}^m$, $\mathcal{L}_{\text{pred}}^a$, and $\mathcal{L}_{\text{pred}}^s$, and the overall $\mathcal{L}_{\text{pred}}$ is redefined as:

$$\mathcal{L}_{\text{pred}} = \mathcal{L}_{\text{pred}}^m + \mathcal{L}_{\text{pred}}^a + \mathcal{L}_{\text{pred}}^s. \tag{12}$$

Let $\bar{Y}^{0,m}$, $\bar{Y}^{0,a}$, and $\bar{Y}^{0,s}$ represent the masked token sequences, generated by masking the tokens identified as difficult by $G_{\phi_{tea}}$ for each level of conditions. Using each condition and masked sequence, $\mathcal{L}_{\text{rec}}$ is redefined as:

$$\mathcal{L}_{\text{rec}} = \sum_{k \in \mathcal{M}} \left[ -\log F_\theta(y_k^0 | \bar{Y}^{0,m}, C^m) - \log F_\theta(y_k^0 | \bar{Y}^{0,a}, [C^m; C^a]) - \log F_\theta(y_k^0 | \bar{Y}^{0,s}, [C^m; C^a; C^s]) \right]. \tag{13}$$

Throughout the overall process, a single model with shared weights is used, rather than training separate models for each of the three types of conditions. This approach serves two purposes. First, it improves efficiency by reducing the number of parameters, making the training process faster and more memory-efficient. Second, weight sharing promotes the learning of consistent and generalizable features across all conditions. Without shared weights, the model may learn disjointed representations for each condition, leading to incoherent or misaligned motion predictions.

### 3.4 INFERENCE: HIERARCHICAL PROCESS FOR MOTION GENERATION

As shown in Fig. 2, the inference process involves two stages. In the first stage, we generate the base-layer token sequence using a hierarchical approach. Starting from an empty token sequence $Y^0(0)$,

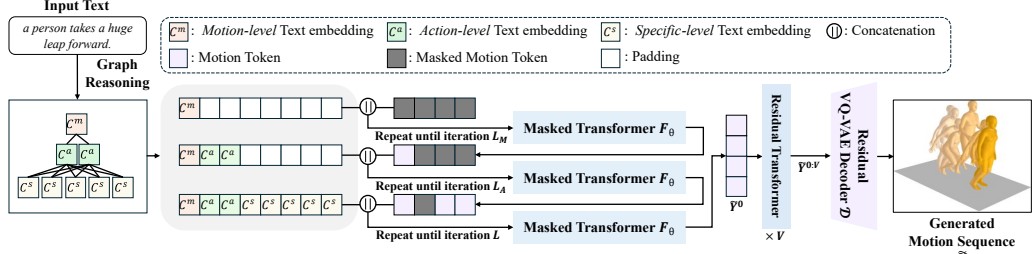

Figure 2: **Overview of the hierarchical inference process for motion generation.** Given an empty motion sequence (with all tokens masked), the input text is transformed into hierarchical semantic features and concatenated with the masked motion tokens. The masked transformer $F_\theta$ is applied iteratively: first utilizing motion-level features for $L_M$ iterations, then adding action-level features until $L_A$ iterations, and finally using all three levels until $L$ iterations. The generated tokens $\tilde{Y}^0$ are then iteratively passed through the pre-trained residual transformer to produce $\tilde{Y}^{0:V}$, which is decoded by VQ-VAE to generate the motion sequence $\tilde{X}$.

where all tokens are masked, the model generates the base-layer token sequence $Y^0 = Y^0(L)$ of length $n$ over $L$ iterations. At each iteration $l$, the masked transformer predicts the probability distribution for each possible token index at the masked positions, which reflects the model's confidence in each prediction. The $\lceil \gamma(\frac{l}{L}) \cdot n \rceil$ tokens with the lowest confidence are masked again. This process repeats until the base-layer tokens are fully generated after $L$ iterations.

During the $L$ iterations, the model is conditioned hierarchically at different levels to progressively refine the motion tokens. This allows the model to capture both general and detailed information from the input text. In the initial iterations, the model is conditioned on the motion level semantic feature, which provides high-level guidance for the overall structure of the motion sequence. As the iterations proceed, the model incorporates the action-level condition, adding more precise information about the actions involved in the motion. Finally, in the later iterations, the specific level condition is applied, introducing finer details about the agents or objects interacting in the scene. Each condition $C(l)$ for the $l$-th iteration is provided as:

$$C(l) = \begin{cases} C^m & \text{if } 1 \le l \le L_M \\ [C^m; C^a] & \text{if } L_M < l \le L_A \\ [C^m; C^a; C^s] & \text{if } L_A < l \le L \end{cases}. \tag{14}$$

where $L_M$ and $L_A$ are the transition points between the levels of conditioning during the $L$ iterations.

After the base-layer tokens are generated through the hierarchical process, the final motion sequence is reconstructed by the pre-trained residual transformer. The residual transformer progressively predicts $V$ token sequences containing residual information, with each sequence refining details lost during quantization. Finally, the predicted residual tokens are combined with the base-layer tokens, which is passed through the VQ-VAE decoder to produce the motion sequence.

## 4 EXPERIMENT

In this section, we present various experimental results for our model, HGM$^3$. Sec. 4.1 covers the overall experimental setup. In Sec. 4.2, we demonstrate the superior quantitative and qualitative results of our model compared to other state-of-the-art methods. Sec. 4.3 presents an ablation analysis and discussion, showing the effectiveness of the components that make up our model.

### 4.1 EXPERIMENTAL SETUP

**Datasets.** We evaluate HGM$^3$ on two widely-used text-to-motion datasets: *HumanML3D* (Guo et al., 2022a) and *KIT Motion-Language (KIT-ML)* (Plappert et al., 2016). **HumanML3D** is currently the largest publicly available dataset for 3D human motion with textual annotations. It contains 14,616 motion sequences sourced from the AMASS (Mahmood et al., 2019) and HumanAct12 (Guo et al., 2020) datasets, paired with 44,970 textual descriptions. Each motion is accompanied by at least three detailed text annotations, with an average description length of approximately 12 words. The motions cover a wide range of activities such as exercising, dancing, and everyday actions. The motion sequences are processed at 20 frames per second (FPS) and have durations between 2 and 10 seconds. **KIT-ML** consists of 3,911 human motion sequences and 6,278 textual descriptions. Each motion is annotated with one to four textual descriptions, with an average of 8 words per description. Both datasets are split into training, validation, and test sets with proportions of 80%, 5%, and 15%, respectively.

Table 1: **Quantitative results on the HumanML3D test set.** All evaluations were replicated 20 times and the average with a 95% confidence interval are reported. (**Bold**: best result, Underlined: second-best)

| Methods | R-Precision ↑ | | | FID ↓ | MM-Dist ↓ | Diversity ↑ | MModality ↑ |
|---|---|---|---|---|---|---|---|
| | Top-1 ↑ | Top-2 ↑ | Top-3 ↑ | | | | |
| Hier | $0.301^{\pm.002}$ | $0.425^{\pm.002}$ | $0.552^{\pm.004}$ | $6.523^{\pm.024}$ | $5.012^{\pm.018}$ | $8.332^{\pm.042}$ | - |
| TEMOS | $0.424^{\pm.002}$ | $0.612^{\pm.003}$ | $0.722^{\pm.002}$ | $3.734^{\pm.028}$ | $3.703^{\pm.008}$ | $8.973^{\pm.071}$ | $0.368^{\pm.018}$ |
| TM2T | $0.424^{\pm.003}$ | $0.618^{\pm.003}$ | $0.729^{\pm.003}$ | $1.501^{\pm.017}$ | $3.467^{\pm.011}$ | $8.589^{\pm.076}$ | $2.424^{\pm.093}$ |
| T2M | $0.455^{\pm.002}$ | $0.636^{\pm.003}$ | $0.736^{\pm.003}$ | $1.087^{\pm.021}$ | $3.347^{\pm.008}$ | $9.175^{\pm.083}$ | $2.219^{\pm.074}$ |
| MDM | $0.320^{\pm.005}$ | $0.498^{\pm.004}$ | $0.611^{\pm.007}$ | $0.544^{\pm.044}$ | $5.566^{\pm.027}$ | $9.559^{\pm.086}$ | $2.799^{\pm.072}$ |
| MotionDiffuse | $0.491^{\pm.001}$ | $0.681^{\pm.002}$ | $0.782^{\pm.002}$ | $0.630^{\pm.011}$ | $3.113^{\pm.001}$ | $9.410^{\pm.049}$ | $1.553^{\pm.042}$ |
| MLD | $0.481^{\pm.003}$ | $0.673^{\pm.002}$ | $0.772^{\pm.002}$ | $0.473^{\pm.013}$ | $3.196^{\pm.010}$ | $9.724^{\pm.082}$ | $2.413^{\pm.079}$ |
| Fg-T2M | $0.492^{\pm.002}$ | $0.683^{\pm.002}$ | $0.783^{\pm.003}$ | $0.243^{\pm.005}$ | $3.109^{\pm.007}$ | $9.278^{\pm.072}$ | $1.614^{\pm.049}$ |
| M2DM | $0.497^{\pm.003}$ | $0.682^{\pm.002}$ | $0.763^{\pm.003}$ | $0.352^{\pm.005}$ | $3.134^{\pm.010}$ | $\mathbf{9.926^{\pm.073}}$ | $\mathbf{3.587^{\pm.072}}$ |
| T2M-GPT | $0.491^{\pm.002}$ | $0.680^{\pm.002}$ | $0.775^{\pm.002}$ | $0.116^{\pm.004}$ | $3.118^{\pm.011}$ | $\underline{9.761^{\pm.081}}$ | $1.856^{\pm.011}$ |
| GraphMotion | $0.504^{\pm.003}$ | $0.699^{\pm.002}$ | $0.785^{\pm.002}$ | $0.116^{\pm.007}$ | $3.070^{\pm.008}$ | $9.692^{\pm.067}$ | $2.766^{\pm.096}$ |
| MMM | $0.515^{\pm.002}$ | $0.708^{\pm.002}$ | $0.804^{\pm.003}$ | $0.089^{\pm.006}$ | $2.926^{\pm.007}$ | $9.577^{\pm.050}$ | $1.226^{\pm.035}$ |
| MoMask | $0.521^{\pm.002}$ | $0.713^{\pm.002}$ | $0.807^{\pm.002}$ | $\underline{0.045^{\pm.002}}$ | $2.958^{\pm.008}$ | - | $1.241^{\pm.040}$ |
| BAMM | $\underline{0.525^{\pm.002}}$ | $\underline{0.720^{\pm.003}}$ | $\underline{0.814^{\pm.003}}$ | $0.055^{\pm.002}$ | $\underline{2.919^{\pm.008}}$ | $9.717^{\pm.089}$ | $1.687^{\pm.051}$ |
| HGM$^3$ (ours) | $\mathbf{0.535^{\pm.002}}$ | $\mathbf{0.726^{\pm.002}}$ | $\mathbf{0.822^{\pm.002}}$ | $\mathbf{0.036^{\pm.002}}$ | $\mathbf{2.904^{\pm.008}}$ | $9.545^{\pm.091}$ | $1.206^{\pm.051}$ |

Table 2: **Quantitative results on the KIT-ML test set.** All evaluations were replicated 20 times and the average with a 95% confidence interval are reported. (**Bold**: best result, Underlined: second-best)

| Methods | R-Precision ↑ | | | FID ↓ | MM-Dist ↓ | Diversity ↑ | MModality ↑ |
|---|---|---|---|---|---|---|---|
| | Top-1 ↑ | Top-2 ↑ | Top-3 ↑ | | | | |
| Hier | $0.255^{\pm.006}$ | $0.432^{\pm.007}$ | $0.531^{\pm.007}$ | $5.203^{\pm.107}$ | $4.986^{\pm.027}$ | $9.563^{\pm.072}$ | - |
| TEMOS | $0.353^{\pm.006}$ | $0.561^{\pm.007}$ | $0.687^{\pm.005}$ | $3.717^{\pm.051}$ | $3.417^{\pm.019}$ | $10.84^{\pm.100}$ | $0.532^{\pm.034}$ |
| TM2T | $0.280^{\pm.005}$ | $0.463^{\pm.006}$ | $0.587^{\pm.005}$ | $3.599^{\pm.153}$ | $4.591^{\pm.026}$ | $9.473^{\pm.117}$ | $3.292^{\pm.081}$ |
| T2M | $0.361^{\pm.006}$ | $0.559^{\pm.007}$ | $0.681^{\pm.007}$ | $3.022^{\pm.107}$ | $3.488^{\pm.028}$ | $10.72^{\pm.145}$ | $2.052^{\pm.107}$ |
| MDM | $0.164^{\pm.004}$ | $0.291^{\pm.004}$ | $0.396^{\pm.004}$ | $0.497^{\pm.021}$ | $9.191^{\pm.022}$ | $10.85^{\pm.109}$ | $1.907^{\pm.214}$ |
| MotionDiffuse | $0.417^{\pm.004}$ | $0.621^{\pm.004}$ | $0.739^{\pm.004}$ | $1.954^{\pm.064}$ | $2.958^{\pm.056}$ | $11.10^{\pm.143}$ | $0.730^{\pm.013}$ |
| MLD | $0.390^{\pm.008}$ | $0.609^{\pm.008}$ | $0.734^{\pm.007}$ | $0.404^{\pm.027}$ | $3.204^{\pm.017}$ | $10.80^{\pm.117}$ | $2.192^{\pm.071}$ |
| Fg-T2M | $0.418^{\pm.005}$ | $0.626^{\pm.004}$ | $0.745^{\pm.004}$ | $0.571^{\pm.047}$ | $3.114^{\pm.015}$ | $10.93^{\pm.083}$ | $1.019^{\pm.029}$ |
| M2DM | $0.416^{\pm.004}$ | $0.628^{\pm.004}$ | $0.743^{\pm.004}$ | $0.515^{\pm.029}$ | $3.015^{\pm.017}$ | $\mathbf{11.417^{\pm.97}}$ | $\underline{3.325^{\pm.37}}$ |
| T2M-GPT | $0.402^{\pm.006}$ | $0.619^{\pm.005}$ | $0.737^{\pm.006}$ | $0.717^{\pm.041}$ | $3.053^{\pm.026}$ | $10.86^{\pm.094}$ | $1.912^{\pm.036}$ |
| GraphMotion | $0.429^{\pm.007}$ | $0.648^{\pm.006}$ | $0.769^{\pm.006}$ | $0.313^{\pm.013}$ | $3.076^{\pm.022}$ | $\underline{11.12^{\pm.135}}$ | $\mathbf{3.627^{\pm.113}}$ |
| MMM | $0.404^{\pm.005}$ | $0.621^{\pm.005}$ | $0.744^{\pm.004}$ | $0.316^{\pm.028}$ | $2.977^{\pm.019}$ | $10.910^{\pm.101}$ | $1.232^{\pm.039}$ |
| MoMask | $0.433^{\pm.007}$ | $0.656^{\pm.005}$ | $0.781^{\pm.005}$ | $0.204^{\pm.011}$ | $2.779^{\pm.022}$ | - | $1.131^{\pm.043}$ |
| BAMM | $\underline{0.438^{\pm.009}}$ | $\underline{0.661^{\pm.009}}$ | $\underline{0.788^{\pm.005}}$ | $\underline{0.183^{\pm.013}}$ | $\underline{2.723^{\pm.026}}$ | $11.008^{\pm.094}$ | $1.609^{\pm.065}$ |
| HGM$^3$ (ours) | $\mathbf{0.444^{\pm.007}}$ | $\mathbf{0.664^{\pm.007}}$ | $\mathbf{0.791^{\pm.006}}$ | $\mathbf{0.176^{\pm.010}}$ | $\mathbf{2.710^{\pm.019}}$ | $10.882^{\pm.081}$ | $1.152^{\pm.041}$ |

**Baselines and evaluation metrics.** We evaluate HGM$^3$ using the baselines referenced in the Sec. 2.1, which include state-of-the-art text-to-motion generation methods, as listed in Tab. 1 and Tab. 2. The evaluation is conducted using five commonly adopted metrics in text-to-motion generation tasks. All metrics use embeddings extracted from pre-trained models following the evaluation protocol from Guo et al. (2022a). (1) **R-Precision** assesses the alignment between input text and generated motion by ranking the Euclidean distances between the motion and 32 text descriptions (1 ground-truth and 31 mismatched). We report Top-1, Top-2, and Top-3 retrieval accuracy. (2) **Frechet Inception Distance (FID)** (Heusel et al., 2017) measures the quality of generated motions by comparing the distribution of their features with real motions. (3) **Multimodal Distance (MM-Dist)** calculates the Euclidean distance between each text feature and the corresponding generated motion feature, evaluating how well the generated motion aligns with the text. (4) **Diversity** computes the average Euclidean distance between randomly sampled pairs of generated motions. (5) **Multimodality (MModality)** measures the model's ability to generate diverse motions from the same text by averaging Euclidean distances between motions generated from a single description.

**Implementation Details.** The residual VQ-VAE contains 6 quantization layers, each with a codebook of 512 codes of 512 dimensions. The downsampling rate $N/n$ of VAE encoder is set to 4 and latent dimension is set to 384. $\beta$ in $\mathcal{L}_{rvq}$ is set to 0.02. We use the ViT-B-32 CLIP model, where the dimension of the text representation is set to 512. All transformers consist of 6 transformer layers with 6 attention heads. For the HTM implementation, $\alpha_0$ and $\alpha_T$ are set to 0 and 0.5, respectively. Our models are trained using the AdamW optimizer for 500 epochs. The learning rate is linearly warmed up to 2e-4 over 2000 iterations. The batch size is set to 512 for training residual VQ-VAE, and 256 for training the masked transformer and the transformer predicting the reconstruction loss.

| HGM³ (ours) | MoMask | T2M-GPT | MLD |
|---|---|---|---|

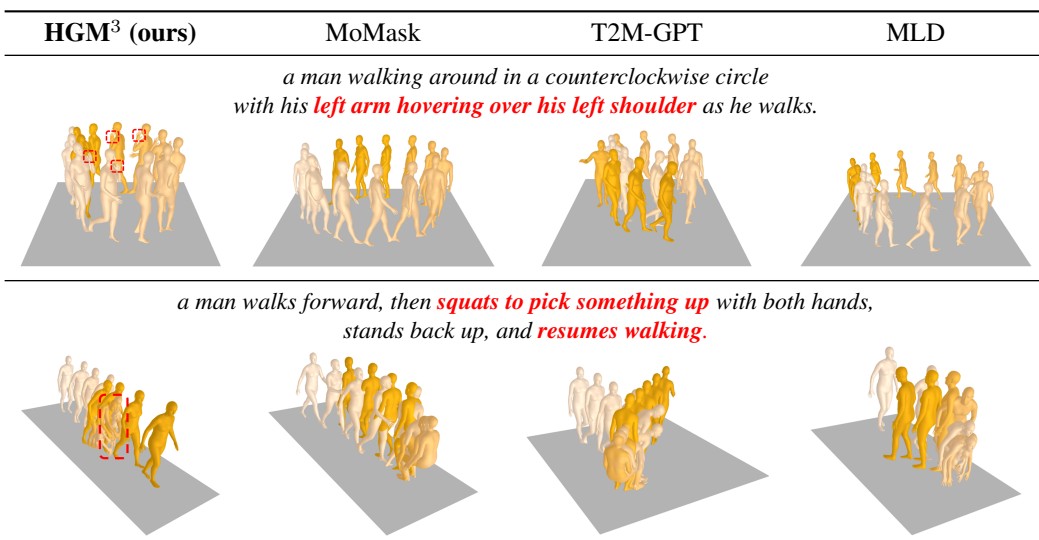

*a man walking around in a counterclockwise circle
with his **left arm hovering over his left shoulder** as he walks.*

*a man walks forward, then **squats to pick something up** with both hands,
stands back up, and **resumes walking**.*

Figure 3: **Qualitative comparisons.** We present the motions generated by various models for two different text prompts, where darker colors indicate later timestamps. Our model not only comprehends the overall semantics of the given text but also captures detailed information effectively compared to other models.

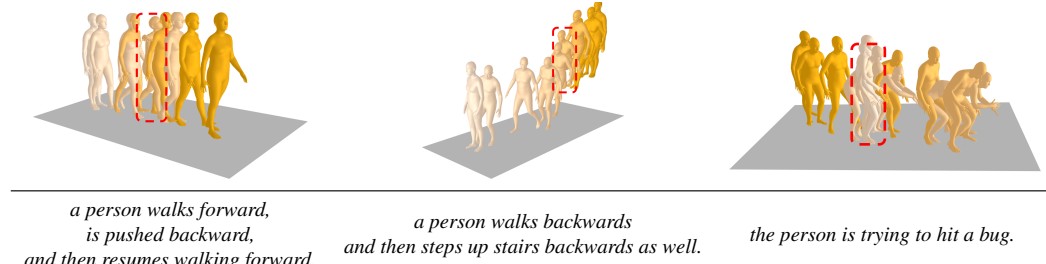

*a person walks forward,
is pushed backward,
and then resumes walking forward.*

*a person walks backwards
and then steps up stairs backwards as well.*

*the person is trying to hit a bug.*

Figure 4: **Visualization of text-to-motion generation results by HGM³.** The red boxes highlight regions in each motion sequence where the HTM model $G_{\phi_{tea}}$ predicts the highest reconstruction loss. These areas correspond to sudden changes or specific, detailed actions, such as being pushed backward, stepping up stairs backward, or attempting to hit a bug. By focusing on these challenging regions during training, the model achieves a high level of consistency between the generated motions and the given textual descriptions.

For training the residual transformer, the batch size is set to 64 for HumanML3D and 32 for KIT-ML. During the inference process, $L_M$, $L_A$, and $L$ are set to 2, 5, and 10, respectively. All our experiments are conducted on a single NVIDIA RTX 6000 Ada Generation GPU.

## 4.2 COMPARISON TO STATE-OF-THE-ART APPROACHES

**Quantitative Results.** The test results for the HumanML3D and KIT-ML datasets are presented in Tab. 1 and Tab. 2, respectively. Following standard practices in text-to-motion generation (Guo et al., 2022a), each experiment was repeated 20 times, and the reported values represent the mean with 95% confidence interval. Our model consistently achieves superior performance compared to recent state-of-the-art methods across multiple metrics, including R-Precision, FID, and MM-Distance. While diversity and MModality are important, they should be viewed as secondary metrics along primary measures, i.e., FID and R-Precision (Guo et al., 2024). HGM³ achieves a superior FID of 0.036, much lower than M2DM's 0.352 on the HumanML3D, indicating higher output quality.

**Qualitative Results.** Fig. 3 presents qualitative comparisons between our model, MoMask, T2M-GPT, and MLD. In the first row, only the motion generated by our model accurately exhibits the left arm hovering over the left shoulder while walking, whereas the others walk in a counterclockwise direction but fail to depict this detail. In the second row, the motions from the other models do not resume walking after standing back up, while our model correctly performs the full sequence. These results demonstrate that our model effectively captures fine-grained details and excels in generating motions that other models struggle to reproduce.

In Fig. 4 we visually demonstrate the validity of HTM, by highlighting the regions where $G_{\phi_{tea}}$ predicts the highest reconstruction loss. These regions typically correspond to parts of the motion involving sudden changes or very specific actions. By focusing the learning process on such regions, our model is able to generate motions that are highly consistent with the given text. For additional qualitative results, including heatmaps illustrating the output of $G_{\phi_{tea}}$, please refer to Appendix C.

### 4.3 ABLATION STUDY

**Text Conditioning and Associated Training and Inference Strategies.** An ablation study is performed to investigate the effects of different types of text conditioning, as well as the use of hierarchical inference and weight sharing among models. The results are presented in Tab. 3 and the experiments are conducted in the same manner as the evaluation in Tab. 1 on the HumanML3D.

First, when providing three types of text conditions, i.e., $[C^m]$, $[C^m; C^a]$, $[C^m; C^a; C^s]$, the performance is higher in all cases compared to using only $[C^m; C^a; C^s]$. We believe that this contributes to our model effectively balancing global and specific information. Moreover, utilizing a hierarchical process during inference yields improved motion generation, suggesting that generating an overall structure first and subsequently refining details is more effective than attempting to produce a specific motion at one shot. Lastly, we found that using a single model with shared weights during training outperforms using three separate models for each condition type. We attribute this to the ability of the unified model to learn more generalizable features across all three types of conditions.

Table 3: Ablation study on text conditioning and associated training and inference strategies. The results are obtained from the HumanML3D experiment. "-" represents that the strategy cannot be implemented, while ✓ and × indicate whether the strategy is executed or not, respectively.

| Text conditioning | Hierarchical inference | Weight shared | R-Precision Top-1 ↑ | FID ↓ |
|---|---|---|---|---|
| $[C^m; C^a; C^s]$ | - | - | $0.523^{\pm.003}$ | $0.051^{\pm.003}$ |
| $[C^m], [C^m; C^a], [C^m; C^a; C^s]$ | × | × | $0.526^{\pm.004}$ | $0.042^{\pm.002}$ |
| $[C^m], [C^m; C^a], [C^m; C^a; C^s]$ | ✓ | × | $0.531^{\pm.002}$ | $0.040^{\pm.002}$ |
| $[C^m], [C^m; C^a], [C^m; C^a; C^s]$ | ✓ | ✓ | $\mathbf{0.535}^{\pm.002}$ | $\mathbf{0.036}^{\pm.002}$ |

**Mask Strategies.** To identify an effective strategy for HTM, different values of $\alpha_0$ and $\alpha_T$ are evaluated on the HumanML3D. As training progresses, the ratio of hard tokens within the masked tokens increases from $\alpha_0$ to $\alpha_T$, whereas the proportion of randomly masked tokens decreases accordingly. Thus, when both $\alpha_0$ and $\alpha_T$ are set to 0, only random masking is applied throughout the training process, representing the easiest strategy. As $\alpha_T$ increases, the proportion of hard tokens in the same epoch increases, leading to a higher level of difficulty. We found that our model achieved the best performance when $\alpha_0$ and $\alpha_T$ are set to 0 and 0.5, respectively, as shown in Tab. 4. This demonstrates that learning which tokens to mask is effective, while maintaining a certain degree of randomness is necessary to prevent the task from becoming overly difficult. In particular, when both values are set to 1, the performance significantly drops, as the model struggles to learn effectively when it is consistently required to predict challenging parts based solely on less meaningful tokens.

Table 4: Ablation study on masking strategies with varying $\alpha_0$ and $\alpha_T$ values. As training progresses, the proportion of hard tokens among the masked tokens increases from $\alpha_0$ to $\alpha_T$.

| Case | Difficulty | Randomness | $\alpha_0$ | $\alpha_T$ | R-Precision Top-1 ↑ | FID ↓ |
|---|---|---|---|---|---|---|
| random | easy | strong | 0 | 0 | $0.529^{\pm.003}$ | $0.044^{\pm.002}$ |
| learn to mask | | | 0 | 0.5 | $\mathbf{0.535}^{\pm.002}$ | $\mathbf{0.036}^{\pm.002}$ |
| learn to mask | ↓ | ↓ | 0 | 1 | $0.533^{\pm.002}$ | $0.041^{\pm.003}$ |
| learn to mask | hard | weak | 1 | 1 | $0.519^{\pm.002}$ | $0.054^{\pm.003}$ |

## 5 CONCLUSION

We propose a novel framework for human motion generation that integrates HTM and a hierarchical semantic text into a generative masked motion model, i.e., HGM³. By selectively focusing on the most challenging motion tokens during training, HTM enhances the model's ability to capture complex dependencies, thereby generating smoother and more coherent motions. Additionally, the hierarchical semantic graph approach effectively organizes text into embeddings at various scales, allowing the model to incorporate multi-layered contextual information for precise and context-aware motion synthesis. Extensive experiments demonstrate that our model achieves state-of-the-art performance on multiple benchmark datasets, significantly improving both motion quality and semantic alignment compared to existing methods. These results emphasize the effectiveness of our approach and provide a potential for applications with realistic motion generation.

REPRODUCIBILITY

To ensure the reproducibility of our work, we present a detailed illustration of the training and inference process of our Hierarchical Generative Masked Motion Model with HTM in Fig. 1 and Fig. 2. Furthermore, we provide the implementation details for HGM[3] and the experimental results replicated 20 times and the average with a $95\%$ confidence interval on HumanML3D and KIT-ML datasets. We will release the full code and setup to facilitate reproducibility of our work.

ACKNOWLEDGEMENTS

This research was supported by RS-2022-II220290 (40%), NRF-2022R1A2C2092336 (30%), IO240508-09825-01 (Samsung Electronics Co., Ltd, 20%), and RS-2019-II191906 (AI Graduate Program at POSTECH, 10%).

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

## A  ARCHITECTURES OF THE RESIDUAL VQ-VAE AND RESIDUAL TRANSFORMER

### A.1  TRAINING SCHEME OF THE RESIDUAL VQ-VAE

We illustrate the training scheme of the Residual VQ-VAE for motion tokenization in Fig. 1, which is described in Section 3.1 of the main manuscript. **1) VQ-VAE Encoder:** The VQ-VAE encoder encodes the input motion sequence $X$, generating a continuous latent feature $Z^0$. This feature is then quantized into discrete latent representations $\hat{Z}^0$ through a vector quantization operation $Q(\cdot)$, **2) Residual Processing:** To effectively capture lost information during quantization of $Z^0$ to $\hat{Z}^0$, a residual learning approach is employed. Specifically, the lost information is retained for every layer and used as input for the next layer, i.e., $Z^{v+1} = Z^v - \hat{Z}^v$. **3) Quantization Layers:** The architecture includes $V$ residual quantization layers. Each layer quantizes the continuous latent features $Z^v$ into discrete features $\hat{Z}^v$. Here, $\hat{Z}^v$ is obtained by mapping each vector in $Z^v$ to the nearest entry in the corresponding learned codebook. **4) VQ-VAE Decoder:** Finally, the aggregated quantized latent features from all layers are fed into the VQ-VAE decoder, which reconstructs the original motion sequence $\hat{X}$.

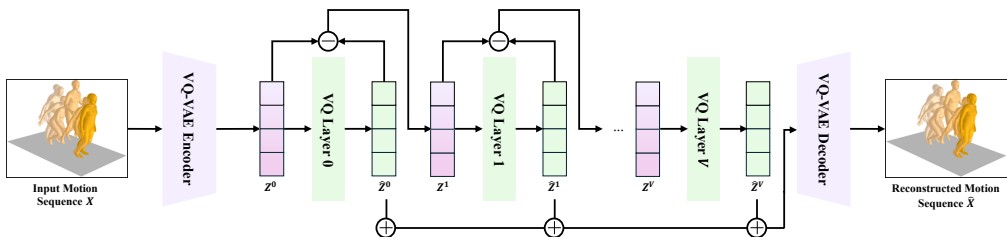

Figure 1:  Training Scheme of the Residual VQ-VAE

### A.2  TRAINING SCHEME OF THE RESIDUAL TRANSFORMER

As shown in Fig. 2, the residual transformer is trained to predict the next layer's token $Y^{i+1}$ as $\tilde{Y}^{i+1}$ based on the tokens from the previous layers $\{Y^0, \ldots, Y^i\}$ and text embedding $c$ obtained from the CLIP encoder. The latent representations from the Residual VQ-VAE, denoted as $\{\hat{Z}^0, \ldots, \hat{Z}^i\}$, are indexed into discrete token sequences $\{Y^0, \ldots, Y^i\}$. Fig. 2 illustrates this single residual transformer, which is applied iteratively across layers. The reconstruction loss is defined as:

$$\mathcal{L}_{\text{res}} = \sum_{i=0}^{V-1} \sum_{j=1}^{n} -\log R_\psi(y_j^{i+1}|Y^{0:i}, c). \tag{1}$$

where $y_j^i$ is the $j$-th token of $Y^i$, and $Y^i$ is obtained by indexing $\hat{Z}^i$.

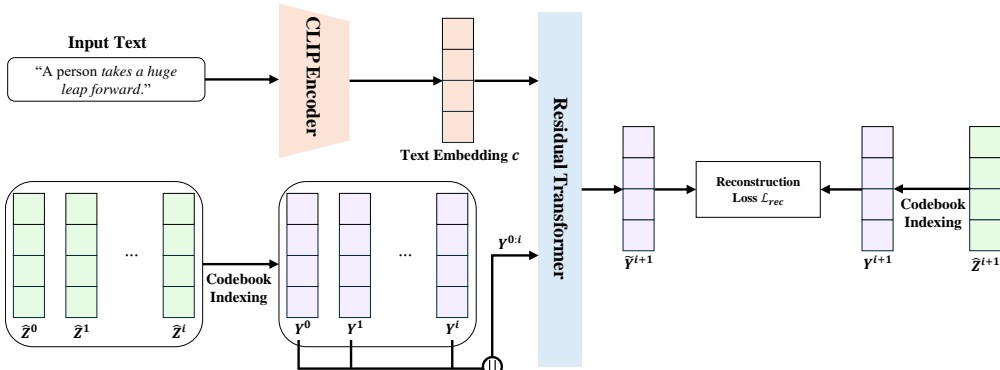

Figure 2:  Training Scheme of the Residual transformer

## A.3 Motion Sequence Generation with Residual VQ-VAE and Residual Transformer

The residual transformer is used to predict the residual layer tokens $\tilde{Y}^1, \cdots, \tilde{Y}^V$ from the generated base layer token $\tilde{Y}^0$ during the motion generation phase outlined in Section 3.4 of the main paper, as illustrated in Fig. 3. This single residual transformer operates iteratively across $V$ layers, with each iteration predicting intermediate motion tokens based on the previous layer's output and the text embedding. As the process progresses through $V$ iterations, the model incrementally incorporates residual information, ensuring that each successive layer captures more detailed aspects of the motion. Ultimately, the outputs from each iteration of the residual transformer, denoted as $\tilde{Y}^{0:V}$, are passed to the VQ-VAE Decoder. This decoder transforms the aggregated motion tokens into a final generated motion sequence $\tilde{X}$, effectively translating the learned representations back into a coherent motion that aligns with the input text description.

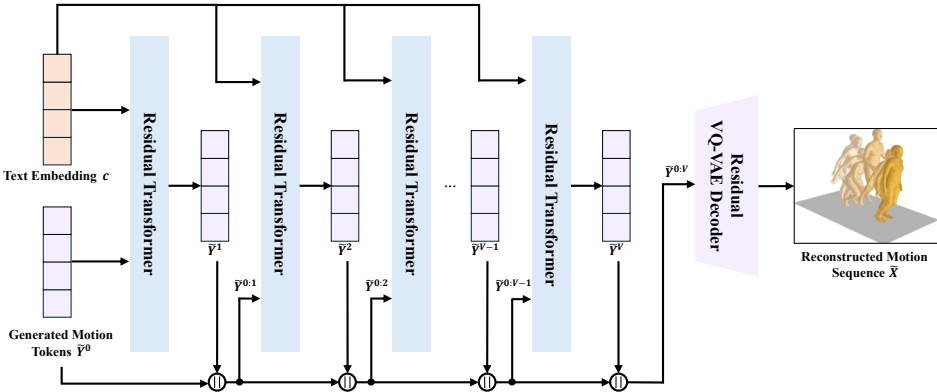

Figure 3: Motion Sequence Generation with Residual VQ-VAE and Residual Transformer

## B Details of Hieralrchical Semantic Graphs

Following (Jin et al., 2024), we employ a semantic role parsing toolkit (Shi & Lin, 2019) to develop Hierarchical Semantic Graphs, enabling us to identify actions along with their attributes and the roles those attributes play within motion descriptions. The parser analyzes the given motion description to extract verbs that signify actions and related attribute phrases, determining the semantic relationships of these phrases concerning the actions. In the graph, the full sentence is represented as a global motion node, while the verbs are modeled as action nodes that connect to the motion node. Attribute phrases are represented as specific nodes associated with the action nodes, with the nature of their connection dictated by the semantic roles of these specifics. The types of nodes and edges in the graph are summarized in Tab. 1.

Table 1: Node and Edge Type Descriptions in Hierarchical Semantic Graphs

|           | Type      | Description                            |
|-----------|-----------|----------------------------------------|
| Node type | Motion    | global motion description               |
|           | Action    | verb                                    |
|           | Specific  | attribute of action                     |
| Edge type | ARG0      | agent                                   |
|           | ARG1      | patient                                 |
|           | ARG2      | instrument, benefactive                 |
|           | ARG3      | start point                             |
|           | ARG4      | end point                               |
|           | ARGM-LOC  | location (where)                        |
|           | ARGM-MNR  | manner (how)                            |
|           | ARGM-TMP  | time (when)                             |
|           | ARGM-DIR  | direction (where to/from)               |
|           | ARGM-ADV  | miscellaneous                           |
|           | ARGM-MA   | motion-action dependencies              |
|           | OTHERS    | other argument types, e.g., action      |

## C   MORE QUALITATIVE RESULTS

Additional qualitative results generated using HumanML3D text prompts are shown in Fig. 4. These results demonstrates that HGM³ successfully captures and reflects a broad spectrum of motion sequences corresponding to various textual descriptions, ensuring a strong semantic alignment between the text and the generated motions.

To further illustrate the model's effectiveness, we include heatmaps of the output of $G_{\phi_{tea}}$ for selected motion sequences in Fig. 5. The heatmaps visualize the predicted reconstruction difficulty across different parts of the motion sequence, with darker regions indicating more challenging areas. This visualization provides deeper insights into how the model identifies and focuses on complex patterns during motion generation.

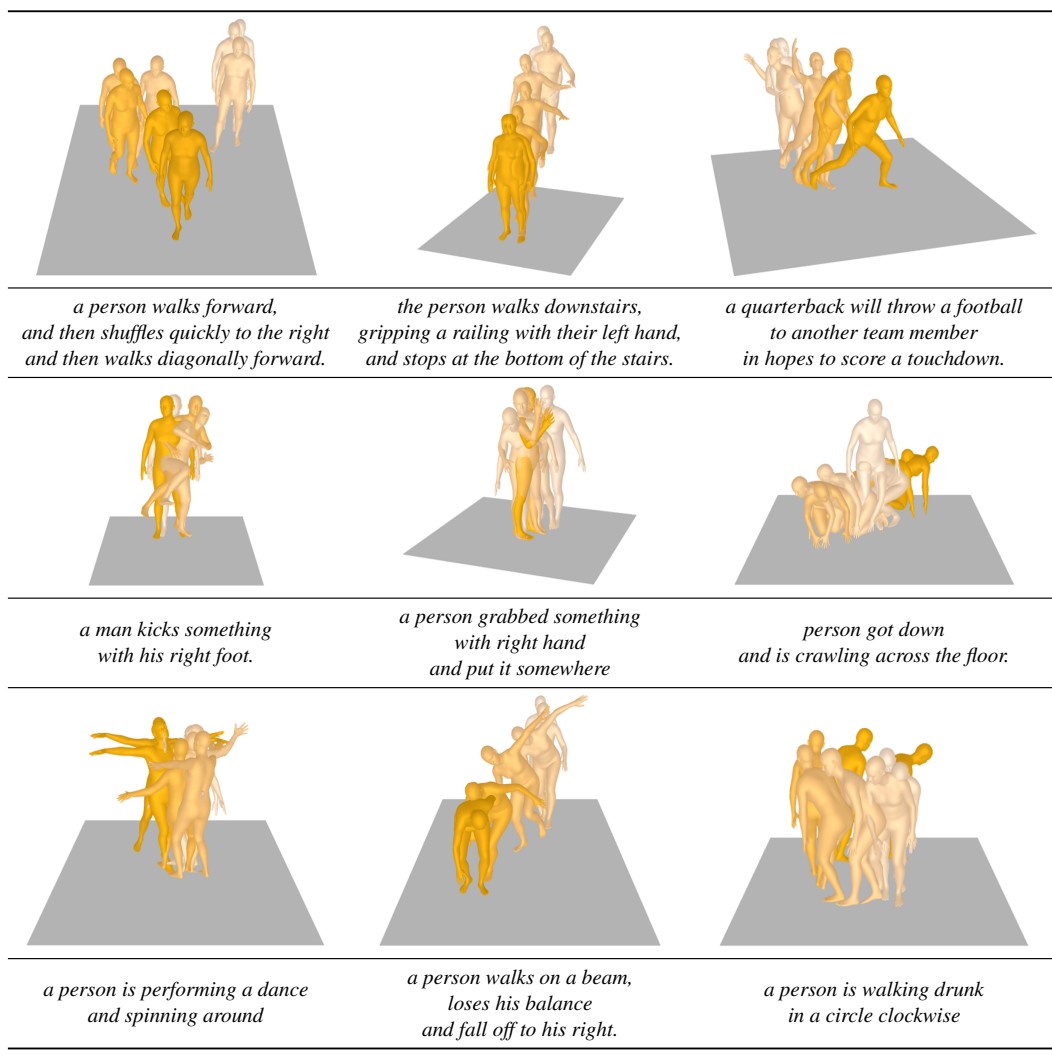

*a person walks forward,*
*and then shuffles quickly to the right*
*and then walks diagonally forward.*

*the person walks downstairs,*
*gripping a railing with their left hand,*
*and stops at the bottom of the stairs.*

*a quarterback will throw a football*
*to another team member*
*in hopes to score a touchdown.*

*a man kicks something*
*with his right foot.*

*a person grabbed something*
*with right hand*
*and put it somewhere*

*person got down*
*and is crawling across the floor.*

*a person is performing a dance*
*and spinning around*

*a person walks on a beam,*
*loses his balance*
*and fall off to his right.*

*a person is walking drunk*
*in a circle clockwise*

Figure 4: **Additional text-to-motion generation results of HGM³.** Darker colors indicate later timestamps.

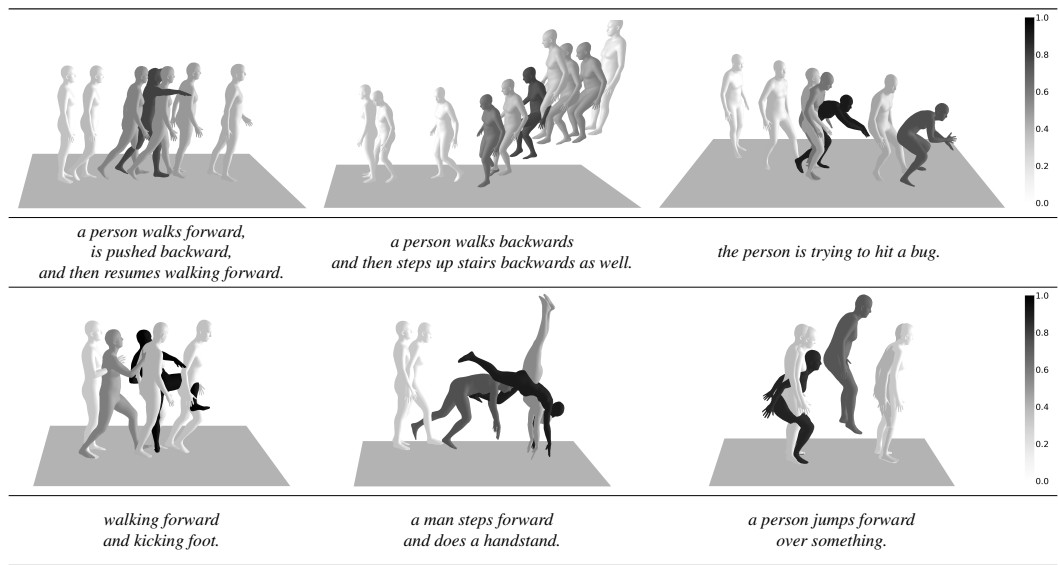

*a person walks forward,*
*is pushed backward,*
*and then resumes walking forward.*

*a person walks backwards*
*and then steps up stairs backwards as well.*

*the person is trying to hit a bug.*

*walking forward*
*and kicking foot.*

*a man steps forward*
*and does a handstand.*

*a person jumps forward*
*over something.*

Figure 5: **Heatmaps of the output of $G_{\phi_{tea}}$ for various motion sequences.** The reconstruction loss is normalized to a range of 0 to 1, with darker colors indicating regions predicted to be more challenging to reconstruct.

# D    ADDITIONAL ABLATION STUDY

An ablation study is conducted to analyze the impact of varying the number of iterations at each stage during the hierarchical inference process according to different overall iteration counts $L$ and transition points $L_M$ and $L_A$. The experiments are performed on HumanML3D, and are reported in Tab. 2. Each condition $C(l)$ for the $l$-th iteration is provided as:

$$C(l) = \begin{cases} C^m & \text{if } 1 \leq l \leq L_M \\ [C^m; C^a] & \text{if } L_M < l \leq L_A \\ [C^m; C^a; C^s] & \text{if } L_A < l \leq L \end{cases} . \tag{2}$$

For the initial setup, we fixed $L$ to 10 and evaluated the performance under different conditions at each stage. When only a single stage is used for inference, providing conditions at the action or specific details improved performance compared to using motion-level condition alone, but the results are still lower than those achieved with a hierarchical approach. In the hierarchical setting, a performance improvement is observed when more iterations are allocated to the final stage compared to the initial stage. This suggests that while all condition types contain motion-level information, specific details can only be effectively utilized in the final stage. As a result, placing more emphasis on the final stage allows the model to better integrate coarse motion structure with fine-grained details, resulting in a more coherent generation. Additionally, we extended the experiments by increasing $L$ to 15 and 20, but found that further increasing the total number of iterations did not significantly impact the model's performance.

Table 2: Ablation study on the number of iterations during inference.

| # of iterations | | | | R-Precision Top-1 ↑ | FID ↓ |
|---|---|---|---|---|---|
| **Total** $L$ | **Motion** $L_M$ | **Action** $(L_A - L_M)$ | **Specific** $(L - L_A)$ | | |
| 10 | 10 | 0 | 0 | $0.522^{\pm.002}$ | $0.041^{\pm.003}$ |
| 10 | 0 | 10 | 0 | $0.525^{\pm.002}$ | $0.046^{\pm.001}$ |
| 10 | 0 | 0 | 10 | $0.527^{\pm.003}$ | $0.043^{\pm.002}$ |
| 10 | 5 | 3 | 2 | $0.532^{\pm.003}$ | $0.038^{\pm.003}$ |
| 10 | 2 | 3 | 5 | $\mathbf{0.535}^{\pm.002}$ | $\mathbf{0.036}^{\pm.002}$ |
| 15 | 3 | 5 | 7 | $0.535^{\pm.003}$ | $0.037^{\pm.002}$ |
| 20 | 4 | 6 | 10 | $0.534^{\pm.002}$ | $0.036^{\pm.002}$ |

