# OpenReview forum: "HGM³: Hierarchical Generative Masked Motion Modeling with Hard Token Mining"
_ICLR.cc/2025/Conference — ICLR 2025 Poster_

### Official Review · Reviewer_3ms7 · 2024-10-27

**Soundness:** 3
**Presentation:** 2
**Contribution:** 2
**Rating:** 6
**Confidence:** 4

**Summary:**

The paper introduces a novel framework with two main components: Hard Token Mining (HTM) and a Hierarchical Generative Masked Motion Model (HGM3). HTM focuses on difficult-to-learn motion areas, while HGM3 represents sentences with different levels of granularity, enabling contextually accurate motion generation by decomposing motion descriptions into hierarchical semantic graphs with three levels: motions, actions, and specifics. This global-to-local structure enables a deep understanding of motion descriptions and provides fine-grained control over motion generation. Experiments show that the propose method out perform state of the art methods for both HumanML3D and KIT dataset.

**Strengths:**

- This work is the first to integrate hierarchical semantic graphs into a masked motion model, offering an innovative approach in this area.
- The literature review effectively outlines the background on text-to-motion, masked motion models, and the challenges in existing methods.
- Comprehensive experiments with detailed comparisons.
- The experiments demonstrate that the proposed method achieves state-of-the-art (SoTA) performance across multiple datasets.

**Weaknesses:**

- It is a bit confusing that $HGM^3$ is used both as the name of a component and as the name of the whole model ($HGM^3 + HTM$).
- $HGM^3$ component:
  - Since the main idea of $HGM^3$ is inspired by GraphMotion [1], it would strengthen the contribution if the authors could demonstrate the differences in applying GAT to a masked model.
  - The $HGM^3$ component is not well described. It is unclear how 1) motions, 2) actions, and 3) specific elements work within the Graph Attention Network.
  - There are no details provided on the "twelve types of edges representing various relationships between nodes" mentioned in Section 3.3.
- The visualization in Figure 4 is unclear. A few more samples (or a video supplement, if easier) may help clarify the results.
- It is unclear if the model truly requires a student-teacher architecture. (I don’t fully understand this part—I’ve also addressed this in the questions section.)


[1] Peng Jin, Yang Wu, Yanbo Fan, Zhongqian Sun, Wei Yang, and Li Yuan. Act as you wish: Finegrained control of motion diffusion model with hierarchical semantic graphs. Neural Information
Processing Systems, 2024.

**Questions:**

- HTM?
  - What is the output of HTM? What is the different between G(student) and F(student) models?
  - What is the motivation of student-teacher architecture? To prevent model collapse? If yes, I think it's better to have ablation study on this.
- $HGM^3$
  - How the integration of Graph Reasoning to masked model is different from GraphMotion?
  - Does Motion-Action-Specific embeddings applied to Residual layers?

---

> ### Author Response · Authors · 2024-11-19
> **Response to Reviewer 3ms7 (Part 1)**
>
> >**W1)** Use of HGM³ as the name of both a component and a whole model (HGM³+HTM).
>
> **A)** Thank you for pointing this out. We agree that using HGM³ for both the framework and a component could lead to confusion, and we are still working on the name of the overall framework. We will post it during the rebuttal period, meanwhile, we would appreciate it if the reviewer can suggest a few candidates.
>
> >**W2-1 / Q2-1)** Demonstrate differences in applying GAT to a masked model from GraphMotion.
>
> **A)** Thank you for your question. While the application of GAT itself remains consistent with GraphMotion, our method differs in how the representations obtained through GAT are utilized. Specifically, we condition three weight-sharing masked models on these representations, which distinguishes our approach from GraphMotion. Additionally, we believe our method offers other novel contributions, which are detailed in the **General Response**.
>
> >**W2-2 / 2-3)** Clarify how motions, actions, and specific elements work within the GAT. / No details on the "twelve types of edges representing various relationships between nodes" in Section 3.3.
>
> **A)** Thank you for your question regarding the hierarchical levels and edge types in our semantic graph. For further details, please refer to Appendix B, where we elaborate on how the hierarchical graph represents sentences through motion, action, and specific levels. Each level provides unique semantic information for text conditioning, guiding the motion generation process. Moreover, as mentioned in line 267, we also describe the twelve types of edges used in the graph in Appendix B, which represent various semantic relationships between nodes. These edges include role-based connections (e.g., agent, patient) and temporal or locational links, which enhance the contextual understanding of the input text. The Graph Attention Network (GAT) uses these levels and relationships to refine the model’s understanding of nuanced text-to-motion alignment, contributing to coherent and contextually accurate generation.
>
> >**W3)** Figure 4 visualization with more samples or a video supplement.
>
> **A)** Thank you for your suggestion. To improve clarity, we have visualized additional examples beyond those shown in Figure 4 of the main paper and included the results in Section C.
>
> For better demonstration, we have included the videos that correspond to the motions in Figure 4 as well as heatmaps of the predicted reconstruction loss distribution (normalized to [0,1] where darker color denotes more challenging motion) in the supplementary.
>
> >**W4)** Necessity of the student-teacher architecture in the model (see questions section).
>
> **A)** We have addressed this question in Q1-2 below. Please refer to that part.

---

> ### Author Response · Authors · 2024-11-19
> **Response to Reviewer 3ms7 (Part 2)**
>
> >**Q1-1)** Output of HTM and difference between G(student) and F(student) models.
>
> **A)** Thank you for your question regarding the HTM and the roles of G(student) and F(student).
>
> **Output of HTM:**
> - The HTM (Hard Token Mining) transformer is designed to identify challenging tokens in the motion sequence by predicting the reconstruction difficulty for each token, rather than directly outputting a mask. The ranking of tokens is then determined by applying an *argsort(⋅)* operation to these predictions in descending order. This ranking process identifies which tokens are most challenging to reconstruct and guides the masking strategy. We have revised lines 240-241 in the manuscript to clarify this distinction (see the text in red).
>
> **Differences Between G(student) and F(student):**
>
> - G(student) serves as the loss predictor, responsible for predicting the reconstruction difficulty for each token in the motion sequence. These predictions are used to compute the loss for the student model, and the model parameters are updated accordingly. G(teacher)​​ is then updated via an Exponential Moving Average (EMA) of the G(student)’s parameters, ensuring that the teacher model gradually incorporates the improvements learned by G(student).
> - F(student), the masked transformer, is trained to reconstruct the challenging tokens identified through the masking process. It receives masked sequences as input and focuses on restoring the original motion tokens by leveraging contextual text embeddings.
>
> By separating these roles, HTM ensures that G(student) focuses on predicting token difficulty, while F(student) specializes in reconstructing difficult tokens. This combination helps the model concentrate on the most challenging parts of the sequence, enhancing its performance in motion generation.
>
> >**Q1-2)** Motivation for student-teacher architecture and need for an ablation study.
>
> **A)** The student-teacher architecture is designed to address the instability that arises when relying solely on the predictions of G(student) during training. While it is possible to directly use the reconstruction loss predicted by G(student), this approach can lead to unstable training due to fluctuating parameters early in training. To mitigate this issue, we introduced a G(teacher) that provides consistent guidance.
>
> G(teacher) is updated using EMA (Exponential Moving Average) of the G(student)'s parameters, ensuring that it evolves gradually and remains stable during training. This stability allows the teacher to offer reliable guidance, enabling F(student) to progressively improve its ability to reconstruct challenging tokens.
>
> To validate the necessity of the EMA mechanism, we conducted an ablation study on the HumanML3D dataset by removing EMA updates and directly updating the teacher model. As shown in the attached table, performance declines without EMA, as the teacher model’s parameters change abruptly, leading to unstable and less effective guidance for F(student).
>
> These findings demonstrate that while the model can operate without EMA updates, the EMA mechanism significantly enhances training stability and overall performance, making it a critical component of our architecture.
>
> | **Methods**       | **Top-1 ↑**          | **Top-2 ↑**          | **Top-3 ↑**          | **FID ↓**           | **MM-Dist ↓**       | **Diversity ↑**     | **MModality ↑**     |
> |--------------------|----------------------|----------------------|----------------------|----------------------|----------------------|----------------------|----------------------|
> | w/o EMA            | 0.524 $^{\pm 0.003}$ | 0.716 $^{\pm 0.002}$ | 0.809 $^{\pm 0.002}$ | 0.049 $^{\pm 0.002}$ | 2.932 $^{\pm 0.009}$ | 9.513 $^{\pm 0.085}$ | 1.228 $^{\pm 0.054}$ |
> | Ours (HGM³)        | 0.535 $^{\pm 0.002}$ | 0.726 $^{\pm 0.002}$ | 0.822 $^{\pm 0.002}$ | 0.036 $^{\pm 0.002}$ | 2.904 $^{\pm 0.008}$ | 9.545 $^{\pm 0.091}$ | 1.206 $^{\pm 0.051}$
>
>
> >**Q2-1)** Differences between integrating Graph Reasoning into a masked model and GraphMotion.
>
> **A)**: We have addressed this question in W2-1. Please refer to that section.
>
> >**Q2-2)** Is Motion-Action-Specific embeddings applied to Residual layers?
>
> **A)** Thank you for your question. Our design is guided by the distinct roles of each transformer in the model. The masked transformer uses hierarchical graph embeddings for fine-grained, multi-level alignment between text and motion, capturing specific nuances in the input. In contrast, the residual transformer refines the motion sequence globally, focusing on coherence and fluidity. For this, we chose generalized text conditioning with CLIP embedding, which was sufficient for global refinement without hierarchical detail.

---

> > ### Comment · Reviewer_3ms7 · 2024-11-26
> > **Official feedback**
> >
> > The rebuttal resolved all my concerns, so I kept my borderline accept rating.

---

### Official Review · Reviewer_oLX4 · 2024-10-31

**Soundness:** 3
**Presentation:** 2
**Contribution:** 2
**Rating:** 6
**Confidence:** 5

**Summary:**

This paper works on the text-to-motion task and proposes two modules to specifically enhance the generative masked modeling-based motion generation. The first contribution is to apply a hard-mining training strategy to replace random masking when training the masked transformer, which facilitates the model to better generate the hard motions. Another contribution is using a graph-based hierarchical text embedding to replace the CLIP text embedding, which provides a better semantic understanding and leads to improved generation quality. Experiments show that the proposed two modules lead to better quantitative and qualitative results compared to the baselines.

**Strengths:**

1. This paper proposes a hard-token mining training strategy that uses additional networks to predict masks of the hard-to-predict tokens. Compared to the random masking strategy used in previous generative masked modeling-based motion generation works, this hard-mining strategy effectively facilitates the models to learn the hard motions.

2. The proposed hierarchical graph-based semantic embedding provides an enhanced text embedding compared to the CLIP text encoder embedding used in previous works. The proposed hierarchical semantic embedding well captures both the fin-grained and global semantics, leading to improved generation results.

3. Experiments show that the proposed two components can effectively enhance the generation quality of generative masked modeling-based methods, and outperforms the baselines both qualitatively and quantitatively.

**Weaknesses:**

1. This paper proposes two components that can specifically enhance generative masked modeling-based motion generation. Although the proposed components are effective, there is not much new knowledge from this paper.  Hard mining is used in the cited reference Hard Patch Mining (HPM) (Wang et al., 2023a) for the image domain and it employs a teacher-student model that transitions from completely random masking to focusing on difficult image patches, in the same spirit as this paper. The hierarchical graph-based embedding is proposed in the reference Act As You Wish, Jin et al. (2024) and proved effective for diffusion-based text-to-motion. This paper applies these two methods to the niche problem of masked modeling-based text-to-motion and achieves enhanced performance.

2. As a motion generation work, this submission does not include any video or animation visualization of motion results. I would strongly suggest including some video results, at least for the sequences presented in Figures 3, 4, and Appendix C.

3. Some technical details regarding text conditioning require further explanation. Line 205 states that the text embedding is from CLIP,  but from other sections I infer that it should be the proposed hierarchical semantic embedding. Moreover, Appendix Figure 2 indicates that the residual transformer uses the text embedding directly from CLIP instead of the hierarchical embedding. What is the reason for not using the hierarchical semantic embedding? I would appreciate additional explanations of the text conditioning used in this work.

I'm on the borderline with this paper given the currently presented manuscript.

**Questions:**

1. The citations in the text of the submitted PDF do not contain clickable links, which makes it hard to match the in-context citation with the reference works. This is not a factor for grading but can make the life of reviewers easier.

---

> ### Author Response · Authors · 2024-11-19
> **Response to Reviewer oLX4**
>
> >**W1)** Not much new knowledge from this paper.
>
> **A)** We appreciate your feedback, which has allowed us to clarify how our work builds on but significantly extends existing methods. While our work draws inspiration from these approaches, it introduces significant innovations to address the unique challenges of masked modeling-based text-to-motion generation:
>
> **Hard Token Mining (HTM):**
> - HTM reimagines the HPM (Wang et al., 2023a) framework by focusing on motion generation, replacing the encoder-decoder structure with a simplified architecture that directly enhances motion token reconstruction.
> - Unlike HPM, which emphasizes invariant representation learning, HTM prioritizes reconstruction quality, introducing input consistency in the teacher-student framework to ensure robust and reliable masking strategies.
>
> **Hierarchical Generative Masked Motion Model (HGM³):**
> - While inspired by GraphMotion(Jin et al., 2024)’s hierarchical graph reasoning, HGM³ introduces unified weight sharing and a single latent space for hierarchical levels. These innovations reduce parameter complexity, improve efficiency, and ensure semantic coherence, addressing limitations in GraphMotion’s separate-level design.
>
> For a detailed explanation of these distinctions, we kindly refer you to the **General Response**, where we discuss the novelty and contributions of HTM and HGM³ in depth.
>
> >**W2)** Video for motion results.
>
> **A)** To address the concern, we have added videos in the supplementary. These include motions from Fig. 3 and Fig. 4 in the main paper, as well as Fig. 4 in Section C, illustrating the diverse and accurate motions generated by our model. The videos clearly demonstrate our model's ability to align motions with the input text while achieving higher precision over baseline methods.
>
> >**W3-1)** Clarify text conditioning in line 205: CLIP embedding vs. hierarchical semantic embedding.
>
> **A)** Thank you for pointing this out. We have clarified this in lines 205 to 208 to prevent any further misunderstanding. As you mentioned, in our model, the masked transformer is conditioned on contextual text embeddings $C$, which represent hierarchical semantic information derived from the input text. This revision has been reflected in the manuscript (in red text) to address this issue.
>
> >**W3-2)** Reason for the residual transformer using CLIP embedding instead of hierarchical embedding.
>
> **A)** Our design choice was based on the distinct roles that each transformer serves in our model. The masked transformer relies on hierarchical graph embeddings to ensure fine-grained, multi-level alignment between the text and the motion, capturing specific nuances in the input text. In contrast, the residual transformer serves as a broad refinement layer, enhancing the coherence and fluidity of the overall motion sequence. For this purpose, we used generalized text conditioning with CLIP embeddings, which was sufficient for global refinement without requiring hierarchical detail.
>
> >**Q1)** Citations in the pdf do not contain clickable links.
>
> **A)** Thank you for bringing this to our attention. We identified one non-clickable link in the main paper and have corrected it to be clickable. Furthermore, we merged the supplementary with the main paper and ensured that all citations are now clickable. If this issue persists, we would greatly appreciate it if you could let us know.

---

> > ### Comment · Reviewer_oLX4 · 2024-11-25
> >
> > Thanks for the detailed response and clarification regarding technical details. The supplementary video results help in understanding the generated motion results.

---

### Official Review · Reviewer_z3vH · 2024-11-01

**Soundness:** 4
**Presentation:** 4
**Contribution:** 1
**Rating:** 6
**Confidence:** 5

**Summary:**

The authors investigate the hard token learning challenge within MoMask's native training scheme and enhance the model's text comprehension by integrating hierarchical textual conditions. Experiments and ablations demonstrate that this implementation improves MoMask's performance.

**Strengths:**

This paper has the following strengths:

- Introduce the Hard Token Mining (HTM) into the motion generation task for the first time and prove its effectiveness.
- Design a Hierarchical Generative Masked Motion Model (HGM$^3$) and use the text conditions with different granularity to enhance the text-motion matching performance.
- Qualitative and quantitative results demonstrate the effectiveness of the proposed method.

**Weaknesses:**

For weaknesses, I have the following comments:

- From an innovation perspective, it’s quite limited. The overall pipeline of the paper is essentially a **replica** of Hard Patches Mining (Wang et al., CVPR 23), merely validated for effectiveness in motion generation.
- As for the proposed Hierarchical Text Graph, while providing more textual information obviously enhances the model's text-motion alignment, a more straightforward approach like using CLIP token-level features would likely be more effective and simpler than this **complex** method (parsing, CLIP, graph, etc.). If the performance of using CLIP token-level features is inferior to the Hierarchical Text Graph, please provide a comparative experiment to demonstrate this.
-  This paper is a well-executed A+B **technical report** with clear and complete experiments, but it offers limited practical value for real-world applications.

**Questions:**

Could Figure 4 show more examples and include **heatmaps of the distribution of the predicted reconstruction loss**, like in Hard Patches Mining (Wang et al., CVPR 23)?

Missing cites:
- MotionGPT: Human Motion as a Foreign Language
- MotionLCM: Real-time Controllable Motion Generation via Latent Consistency Model
- StableMoFusion: Towards Robust and Efficient Diffusion-based Motion Generation Framework
- Mofusion: A framework for denoising-diffusion-based motion synthesis

---

> ### Author Response · Authors · 2024-11-19
> **Response to Reviewer z3vH (Part 1)**
>
> >**W1)** Limited Innovation; pipeline resembles Hard Patches Mining.
>
> **A)** We humbly disagree with the claim that our approach is a replica of HPM (Wang et al., 2023a), and we appreciate the reviewer for a chance to clarify the contribution of our work further. While HTM was inspired by HPM’s targeted masking concept, it has been fundamentally re-designed to address the unique challenges of motion generation:
>
> - **Architectural Differences:** HTM replaces HPM’s encoder-decoder structure with a streamlined design comprising two transformers—a masked transformer (reconstructor) and a loss predictor (HTM transformer)—tailored for motion token reconstruction.
> - **Purpose:** Unlike HPM, which focuses on representation learning for invariant feature extraction, HTM directly prioritizes reconstruction quality by selectively masking challenging motion tokens.
> - **Input Consistency:** HTM resolves the input mismatch issue in HPM by using consistent unmasked token sequences for both teacher and student models, ensuring reliable and robust masking strategies.
>
> We believe these distinctions demonstrate HTM’s novel contributions to the field. For additional details, we kindly refer you to the **General Response**, where we discuss HTM’s innovations in depth.
>
> >**W2** Comparison of Hierarchical Text Graph and CLIP token-level features.
>
> **A)** We appreciate the reviewer’s insights, which is a setting that we totally agree with. We conducted an experiment  on HumanML3D dataset using the entire set of CLIP output tokens in a straightforward manner as conditioning features for text-to-motion alignment, and the results showed a noticeable decline in performance compared to the Hierarchical Text Graph approach. (See the table below.) We believe this is due to several important distinctions between the two methods:
>
> - **Hierarchical Structure with Multi-Level Filtering:**
> Our Hierarchical Text Graph organizes text into meaningful levels—motion, action, and specifics—enabling the model to prioritize and leverage motion-relevant attributes. This global-to-local structure allows the model to better understand the text by capturing semantic information at varying levels of granularity to guide motion generation. In contrast, using CLIP tokens in a straightforward manner often introduces noise, diluting the focus on motion-specific features. Our focused hierarchical approach played a crucial role in achieving precise alignment between text and motion, resulting in high-quality motion generation.
>
> - **Relational and Contextual Encoding with GAT:**
> Our model uses a Graph Attention Network (GAT) to facilitate information exchange between nodes, allowing each node to refine its understanding based on relationships across semantic levels. This relational encoding was essential for capturing dependencies in motion data, as it enables the model to understand temporal and contextual relationships in the text. CLIP tokens alone, in contrast, do not support this dynamic interaction, which leads to poorer alignment and coherence in the generated motions.
>
> Given these observations, we believe that the Hierarchical Text Graph with GAT provides essential structure and contextual interaction that significantly enhances text-to-motion alignment. These factors are not captured by the straightforward use of CLIP token-level features, as reflected in the performance disparity observed in our experiment.
>
> | **Methods**               | **Top-1 ↑**          | **Top-2 ↑**          | **Top-3 ↑**          | **FID ↓**           | **MM-Dist ↓**       | **Diversity ↑**     | **MModality ↑**     |
> |----------------------------|----------------------|----------------------|----------------------|----------------------|----------------------|----------------------|----------------------|
> | CLIP token-level features  | 0.519 $^{\pm 0.003}$ | 0.709 $^{\pm 0.002}$ | 0.808 $^{\pm 0.003}$ | 0.076 $^{\pm 0.003}$ | 2.937 $^{\pm 0.013}$ | 9.510 $^{\pm 0.078}$ | 1.136 $^{\pm 0.045}$ |
> | Ours (HGM³)               | 0.535 $^{\pm 0.002}$ | 0.726 $^{\pm 0.002}$ | 0.822 $^{\pm 0.002}$ | 0.036 $^{\pm 0.002}$ | 2.904 $^{\pm 0.008}$ | 9.545 $^{\pm 0.091}$ | 1.206 $^{\pm 0.051}$ |
>
> >**W3)** Well-executed A+B technical report, but limited practical value for real-world applications.
>
> **A)** We believe that our paper offers more than technical contributions, as it demonstrates significant advancements with practical applicability. Our proposed method not only improves the alignment between text and motion but also generates high-quality, realistic motions, as evidenced by our qualitative results. These results showcase motions that are well-suited for real-world applications such as animation, virtual reality, and gaming. To further illustrate the practicality of our approach, we have added supplementary video materials, which include examples of generated motions. We kindly invite you to explore these materials for further insights into the applicability of our work.

---

> ### Author Response · Authors · 2024-11-19
> **Response to Reviewer z3vH (Part 2)**
>
> >**Q1)** Include more examples and heatmaps in Figure 4.
>
> **A)** In addition to the motions shown in Figure 4 of the main paper, we have visualized heatmaps of the predicted reconstruction loss distribution for three additional motions and reported the results in Section C.
>
> The predicted reconstruction loss values are normalized to a range of 0 to 1, with lightness values varying from 0.0 (white) to 1.0 (black) to represent the heatmaps. Darker colors indicate regions predicted to be more challenging to reconstruct. We would greatly appreciate it if you could review the revised paper for the detailed results. As expected, the motions with high variations were highlighted in dark.
>
> >**Q2)** Missing cites
>
> **A)** We have incorporated discussions of the suggested papers into the related work section. Please refer to the revised text in lines 94 and 97.

---

> ### Comment · Reviewer_z3vH · 2024-11-26
> **To Authors**
>
> One last question: StableMoFusion uses token-level CLIP features, and the R-Precision score is very high (i.e., 0.553), which is quite different from the results in your experiments. Could you explain the possible reasons for this discrepancy?

---

> > ### Author Response · Authors · 2024-11-27
> > **Response to Reviewer z3vH**
> >
> > Thank you for your question regarding the R-Precision discrepancy between StableMoFusion (Huang et al., 2024) and our experiment. We believe the difference arises from factors beyond the use of CLIP token-level features.
> >
> > **1) CLIP Token-Level Features in StableMoFusion and MotionDiffuse (Zhang et al., 2024):**
> >
> > - StableMoFusion adopts the same CLIP embedding strategy as MotionDiffuse, as indicated by its text encoding setup:
> > *“For text encoder, we leverage pre-trained CLIP token embeddings, augmenting them with four additional transformer encoder layers, the same as MotionDiffuse, with a latent text dimension of 256.”*
> >
> > - However, MotionDiffuse reports a significantly lower R-Precision (Top-1) of 0.491, which suggests that CLIP token-level embeddings alone are not sufficient to achieve the high R-Precision observed in StableMoFusion.
> >
> > - This indicates that the architecture modifications or loss design in StableMoFusion likely contribute more significantly to its R-Precision performance, rather than the embedding method itself.
> >
> > **2) Masked Models and Superior FID Performance:**
> >
> > - Recent advancements in masked modeling have led to significant improvements in FID, surpassing diffusion-based models. For instance, our method achieves an FID of 0.036, which is substantially lower than StableMoFusion’s FID of 0.098.
> >
> > - This difference emphasizes the effectiveness of our approach in generating high-quality motions that closely align with real motion distributions.

---

### Official Review · Reviewer_RKiq · 2024-11-03

**Soundness:** 2
**Presentation:** 3
**Contribution:** 2
**Rating:** 6
**Confidence:** 4

**Summary:**

This work suggests a text-to-motion generation model, using the combination of three existing techniques:
1. Masked VQ-VAE in the motion domain:
    1. Hierarchical (MoMask by Guo et al., 2024)
    2. Masking based on confidence (Pinyoanuntapong et al. 2024b)
2. Hard token mining (HTM) in the imaging domain (Wang et al., 2023a)
3. hierarchical semantic graph representation in the language domain (Shi & Lin, 2019) + Graph Attention Network (GAT) (Velickovic et al., 2018)

------------------------------------
**Post rebuttal comment:**

Following the rebuttal, I am raising my score toward acceptance.

**Strengths:**

- Impressive and thoughtful choice of SOTA works.
- Clear writing (mostly).
- Reproducibility: implementation details are given and, more importantly, the authors plan to release their full code and setup.

**Weaknesses:**

**Major weaknesses**

- Novelty: This work combines existing SOTA works (see "Summary" above). While this work demonstrates solid engineering execution in integrating existing techniques, its novel contribution to the field is questionable. Please discuss any new theoretical insights or algorithmic innovations beyond the integration of existing techniques.
- Technical soundness: How is the loss of the residual transformer (L_res) incorporated into the overall network loss? Can you clarify the summation range in Eq. 1 of Sec. A.2? I believe it should be from i=1 to V. Could you provide a diagram showing the architectural flow between the masked and residual transformers?"
- Qualitative results: To fully assess the quality and naturalness of the generated motions, I recommend including a supplementary video. While the paper includes qualitative figures, they cannot be validated for dynamic motion artifacts such as jitteriness and foot sliding.
- Quantitative results: For most metrics, results are only marginally better and sometimes marginally worse. I would call that comparable. There is, however, a notable improvement for HML3D FID.

**More weaknesses**

- Some technical descriptions need more clarification or need to be corrected (see "Questions" below).
- I suggest concatenating the supp to the main paper to allow mutual references and better reader experience. This is allowed in ICLR.

**Questions:**

Questions and Comments:

- Masked and residual transformers: Do they predict all tokens together (i.e., predict $\mathcal{M}$ tokens for the masked transformer and $n$ tokens for the residual one)? Or, are they causal? (by causal I mean that for the residual transformer, predict tokens according to their temporal order where each prediction is conditioned on the previously predicted tokens; for the masked transformer it probably means conditioning masked tokens on those already predicted). If the prediction is causal, then a prediction of an $n$ length motion requires an n-step loop, hence each iteration in Sec. 3.4 has an internal loop within it. Please explain.
- L209.5 Eq. 4): Rephrase. it seems you wanted to describe k and M, but described M only.
- L220 (Eq. 5):  The L_pred objective focuses on ranking losses by relative magnitude rather than exact values. Therefore, $\hat{\ell}$ (and G) should be interpreted as predicting the values with the topological ordering of token losses, not as estimating the precise reconstruction loss.
- L249: "At each step": do you mean "at each epoch"? The index k is related to epochs (L242), not to steps.
- L 249: Does it mean $\mathcal{M} = \gamma(\tau_k)\cdot n$ ? I don't see anywhere in the paper how $\mathcal{M}$ is defined.
- L339: where is the probability distribution taken from?
- Sec. A.2, A.3: Fig. 2 depicts the residual transformer in one block, in blue Fig. 3 depicts the same blue block multiple times, but relates to it as sub-blocks of the residual transformer from Fig. 2. This is misleading. Please clarify the differences and similarities between the sub-blocks and make their notation more consistent.

---

> ### Author Response · Authors · 2024-11-19
> **Response to Reviewer RKiq (Part 1)**
>
> >**W1)**  Discuss any new theoretical insights or algorithmic innovations beyond the integration of existing techniques.
>
> **A)** Thank you for your thoughtful feedback regarding the novelty of our contributions. While our work draws inspiration from existing techniques, it extends these methods with innovations tailored specifically for motion generation tasks.
>
> - Hard Token Mining (HTM): Our HTM approach diverges from Hard Patch Mining (HPM) (Wang et al., 2023a) by focusing on motion token sequence reconstruction and eliminates the encoder-decoder structure, and HTM ensures input consistency in the student-teacher framework.
> - Hierarchical Generative Masked Motion Model (HGM³): Beyond hierarchical semantic graphs, our model introduces unified weight sharing across hierarchical levels and operates within a single latent space, significantly improving efficiency, scalability, and coherence compared to prior methods.
>
> For further details, please refer to the **General Response** where we provide an in-depth explanation of how HTM and HGM³ introduce theoretical and architectural innovations that extend beyond existing methods.
>
>
>
> >**W2-1)** How is $L_{res}$ incorporated into the overall network loss?
>
> **A)** As shown in Eq. 1 in Section A.2, the residual transformer is a model that is trained to predict $Y^{i+1}$ given token sequences $Y^0$ to $Y^i$ obtained from the residual VQ-VAE. As it is trained separately from the masked transformer, therefore, $L_{res}$​ is not included in the overall network loss $L$ (which is composed of $L_{rec}$​ and $L_{pred}​$). During the inference process, this separately trained residual transformer sequentially predicts $Y^1$ to $Y^V$ from the masked transformer output $\tilde{Y}^0$ , as described in Section A.3. $\tilde{Y}^{0:V}$ is then decoded into the motion sequence via the VQ-VAE decoder.
>
>
>
> >**W2-2)** Clarification of the summation range in Eq. 1 of Sec. A.2 ($i=1$ to $V$).
>
> **A)** Regarding Eq. 1 in Section A.2, since we need to calculate the loss starting from predicting $Y^1$ given $Y^0$ up to predicting $Y^V$ given $Y^{0:V-1}$, it is correct to define the summation range as $i = 0$ to $V-1$.
>
>
> >**W2-3)** Diagram of flow between masked and residual transformers.
>
> **A)** We originally had one in the Supplementary in Section A.3, which may not have been clear. We have revised the figure and its description in Section A.3 for better understanding of our model. Additionally, we have updated Figure 2 in the main paper to better illustrate the flow between these two components.
>
>
> >**W3)** Including supplementary video for qualitative results.
>
> **A)** We have included and uploaded video materials in the supplementary to show the motions presented in Fig. 3 and Fig. 4 in main paper, and Fig. 4 in Section C. These videos demonstrate how our model generates diverse motions accurately aligned with the input text. Moreover, the videos highlight that our model produces motions with higher precision compared to those generated by other baseline models.
>
>
> >**W4)** Comparable quantitative results; notable improvement in HumanML3D FID.
>
> **A)** We appreciate the reviewer for recognizing the improved performance of our model in FID on the HumanML3D dataset. FID is a primary metric for evaluating the quality of generated motion, and this result demonstrates that our model captures a high level of realism and detail. Notably, given that the HumanML3D dataset is significantly larger than the KIT-ML dataset, we believe this suggests a potential that our method tends to perform better with more data.
>
> Moreover, as mentioned in Section 4.2, our analysis is focused more on the FID and R-precision and the minor drops observed in diversity and MModality are considered secondary. Placing too much emphasis on multimodality and diversity could lead to outputs that lose relevance or coherence with the input text. Thus, we believed that ensuring output quality and maintaining alignment with the text was more important than simply generating diverse outputs. Our study focuses on maintaining this balance while improving the primary metrics of FID, R-Precision, and Multimodal Distance.
>
> Regarding the marginal improvement in other metrics, incremental improvements between SOTA models are common in existing research. Therefore, we believe the performance gains achieved by the HGM³ over prior SOTA models are by no means small and represent meaningful evidence of the effectiveness of our new method.
>
>
>
> >**W5)** Clarify or correct some technical descriptions (see "Questions" below).
>
> **A)** We have addressed this in the questions section below. Please refer to that section.
>
>
> >**W6)** Concatenating supplementary to the main paper
>
> **A)** Thank you for the helpful suggestion. We have connected the main paper with the supplementary material and re-uploaded it.

---

> ### Author Response · Authors · 2024-11-19
> **Response to Reviewer RKiq (Part 2)**
>
> >**Q1)** Token prediction in masked and residual transformers.
>
> **A)** As mentioned in our answer to W2-1, the masked transformer and residual transformer are trained separately. During inference, an empty motion sequence is given with all $n$ tokens masked, and the masked transformer predicts all tokens over $L$ iterations. This prediction results in $\tilde{Y}^0$, which, as described in Section A.3, is then fed as an  input to the residual transformer. The residual transformer then sequentially predicts $\tilde{Y}^1$ to $\tilde{Y}^V$.
>
> >**Q2)** Clarify Eq. 4: description of $k$ and $\mathcal{M}$.
>
> **A)** Thank you for pointing this out. We have revised the text to clarify both $k$ and $\mathcal{M}$. Specifically, the definition of $\mathcal{M}$, which denotes the set of indices for all masked tokens, has been added in line 204. Additionally, we have included an explanation of $k$ in line 210, which represents the index of each masked token.
>
> We also realized that the $k$ used here could potentially cause confusion with the $k$ later used to denote epochs in the manuscript. To address this, we have updated the notation for epochs to $t$, ensuring a clear distinction between the index $k$ of masked tokens.
>
>
> >**Q3)** L220 (Eq. 5): $\hat{\ell}$ (and $G$) as predicting topological order of token losses.
>
> **A)** Thank you for pointing this out. To prevent any misunderstanding, we have revised the manuscript to clarify that $\hat{\ell}$ and $G$ are intended to predict the relative ordering of token losses rather than their exact reconstruction values. Specifically, we have modified the text in line 201 from 'predicting the reconstruction loss' to 'predicting the relative ordering of reconstruction losses'. Similar adjustments are made in lines 223 and 240 to ensure that the objective of $L_{pred}$​ as a ranking mechanism is clearly conveyed.
>
> >**Q4)** L249: Clarify if "at each step" refers to epochs ($k$).
>
> **A)** Thank you for your observation. 'At each step' in line 249 does indeed refer to 'epoch.' We have revised the manuscript to say 'at each epoch' to prevent any confusion.
>
> >**Q5)** L 249: Clarify if $\mathcal{M}=\gamma(\tau_k)\cdot n$ and define $\mathcal{M}$.
>
> **A)** We had previously defined $\mathcal{M}$ in line 209.5, but we apologize if there was any confusion. To improve clarity, we have now revised the manuscript to specify this definition at line 204. Additionally, regarding your question about $\mathcal{M} = \gamma(\tau_k) \cdot n$ (which has now been updated to $\gamma(\tau_t) \cdot n$ in the revised text), this expression does not define $\mathcal{M}$ directly. Instead, $\gamma(\tau_t) \cdot n$ refers to the total number of tokens selected for masking, whereas $\mathcal{M}$ denotes the specific indices of these selected tokens in the motion sequence. We would like to clarify that $\mathcal{M}$ is recalculated at each epoch $t$, as the masking strategy is updated dynamically based on the model's predictions. This has been further clarified in line 204.
>
> >**Q6)** L339: where is the probability distribution taken from?
>
> **A)** The probability distribution mentioned here is generated by the masked transformer, which, at each iteration $l$, uses a softmax function to produce the likelihood of each possible token index for the masked positions. This probability distribution reflects the model's confidence in its predictions for each token, with the tokens having the lowest confidence subsequently masked again in the following iteration. Please refer to the revised manuscript for the updated explanation in line 338.
>
> >**Q7)** Sec. A.2, A.3: Differences and similarities between residual transformer blocks in Fig. 2 and Fig. 3.
>
> **A)** In Sections A.2 and A.3, the blue block in both Fig. 2 and Fig. 3 represents a single residual transformer. During training, the residual transformer takes $Y^{0:i}$ as input, where the tokens are embedded and summed to predict $Y^{i+1}$. Fig. 2 is intended to illustrate this process. In both training and inference, the same residual transformer consistently predicts the next token sequence layer, regardless of the specific input index $i$. This means that, during inference, a single residual transformer is applied iteratively across different inputs. Fig. 3 reflects this by depicting multiple applications of the same residual transformer to illustrate its repeated use, not multiple sub-blocks. Please refer to the revised manuscript for the updated explanation.

---

> ### Comment · Reviewer_RKiq · 2024-11-22
> **Addressing the Authors' Rebuttal**
>
> I appreciate the thorough, impressively written, rebuttal.
>
> While most of my concerns have been addressed, one key issue remains unresolved.
>
> The newly added video supplementary, particularly the files under the "Additional videos from ours" folder, exhibit low-quality motions. Most of them have severe floating artifacts, some depict unacceptably jittery motions, and some show unnatural motions.
> The quality of these motions is much lower than in many prior works.
>
> Here are three (out of many) examples:
> - The file "a person grabbed something with right hand and put it somewhere" shows severe floating artifacts.
> - The file "a person is performing a dance and spinning around" shows an unnatural motion.
> - The file "a quarterback will throw a football to another team member in hopes to score a touchdown" shows jittery head motions.
>
> The most important outcome in a work is the quality of synthesized motions. With low-quality results, this work cannot be used, no matter how well it is written and what advanced building blocks it uses.
>
> Given the aforementioned considerations, I believe a rating of 5 remains appropriate.
>
> I urge the authors to find the cause of the exhibited artifacts, fix their model, and re-submit their work to another venue.

---

> ### Author Response · Authors · 2024-11-22
> **Response to Reviewer RKiq**
>
> We sincerely thank you for your detailed observations regarding the supplementary videos. While we appreciate your thorough review, we humbly disagree with the assertion that the quality of our generated motions is "much lower than in many prior works." Below, we address your concerns.
>
> ---
>
> **Comparative Motion Quality**
>
> While some floating and jittery artifacts are present, they are not unique to our model. Such issues arise naturally from the **SMPL rendering process** when no post-rendering corrections are applied. For fair comparisons, we intentionally refrained from applying corrective techniques. These corrections include:
>
> - **Foot Contact Stabilization:** Ensures feet remain grounded during contact phases, eliminating floating effects.
>
> - **Temporal Smoothing:** Reduces jittering in movements by filtering sudden transitions between frames.
>
> - **Head Alignment Regularization:** Minimizes head jitter by constraining rapid orientation changes.
>
> As examples:
>
> - The **MMM project page** (Pinyoanuntapong et al., 2024b) shows similar jittering or floating artifacts in uncorrected outputs.
>
> - The **MoMask project page** (Guo et al., 2024) displays outputs where such artifacts appear corrected, but our experiments with uncorrected MoMask outputs reveal similar issues, such as floating and jittering.
>
> These examples demonstrate that such artifacts are common in raw SMPL-rendered outputs across leading methods.
>
>
> **Instruction Alignment and Comparative Performance**
>
> To address your concerns, we have created comparison videos with outputs from other models. **These videos are uploaded in the supplementary material,** showing the three motions you highlighted. The videos clearly illustrate the following:
>
> - While our model exhibits artifacts, they are observed in all other models without post-rendering corrections.
>
> - Despite the artifacts, our model adheres to all instructions for motions containing detailed descriptions. In contrast, other models not only exhibit artifacts but also fail to align their motions with the given instructions, producing outputs that deviate from the semantics of the textual input.
>
> Based on these observations, we respectfully disagree with the notion that our model's performance is inferior. Our method demonstrates superior instruction adherence, which we consider a critical factor for evaluating motion generation models.
>
>
>
> **Commitment to Future Improvements**
>
> The primary goal of our paper is to generate motion that accurately reflects the details and nuances of the input text, rather than focusing on artifact correction methods. To ensure fair comparisons with other models, we deliberately refrained from applying post-processing techniques, such as temporal smoothing or foot contact stabilization, which could artificially enhance visual quality.
> However, we understand the importance of producing high-quality rendered outputs for broader applications. To address this, we plan to apply the post-rendering correction process as part of our project page. This will reduce artifacts such as jittering and floating motions, while maintaining the generated motion's alignment with the input text.

---

> > ### Comment · Reviewer_RKiq · 2024-11-30
> > **Response to authors**
> >
> > Thank you for the additional video and your thorough answers. I appreciate the extra effort you put into providing detailed and helpful explanations.
> > My concerns were answered and I am raising my score towards acceptance.

---

> > > ### Author Response · Authors · 2024-12-01
> > >
> > > We appreciate the reviewer for reconsidering the evaluation.

---

### Author Response · Authors · 2024-11-19
**General Response to all Reviewers**

Dear Reviewers,

We sincerely thank the reviewers for their thoughtful and constructive feedback. We appreciate the recognition of our contributions, practical value, and experimental rigor. Notably, the reviewers highlighted:

**Innovative Approach:**
- *“...the first to integrate hierarchical semantic graphs into a masked motion model, offering an innovative approach in this area.”* (**Reviewer 3ms7**)
- *“Introduce the Hard Token Mining (HTM) into the motion generation task for the first time and prove its effectiveness.”* (**Reviewer z3vH**)

**Strong Empirical Performance and Comprehensive Experiments:**
- *“...outperforms the baselines both qualitatively and quantitatively.”* (**Reviewer oLX4**)
- *“...state-of-the-art (SoTA) performance across multiple datasets.”* (**Reviewer 3ms7**)
- *“Qualitative and quantitative results demonstrate the effectiveness of the proposed method.”* (**Reviewer z3vH**)
- *“Comprehensive experiments with detailed comparisons.”* (**Reviewer 3ms7**)

While we are grateful for the positive feedback, we also seriously take the reviewers’ concerns especially on the novelty of our work. Below, we provide a detailed response to address these points, and revised texts are given in red in the newly attached manuscript.

We truly hope that our rebuttal clarifies all the critical issues raised by the reviewers and helps the reviewers with a better understanding of our work, possibly leading to higher chances for the acceptance of this paper. We are open for any discussion during the rebuttal period.

---

### Clarifying Novelty: addressing Reviewers’ common concerns

**Hard Token Mining (HTM) vs. Hard Patch Mining (HPM) (Wang et al., 2023a)**

While HTM draws inspiration from the targeted masking approach of HPM, it has been fundamentally redesigned to address the unique challenges of motion data. The key differences are as follows:

1) **Purpose and Architecture**

In HPM, the primary goal is to learn novel representation by training an encoder-decoder architecture to produce mask-invariant features from the encoder with two separate decoders (i.e., reconstructor and loss predictor). In contrast, our HTM aims to enhance reconstruction quality rather than representation learning. By focusing on improving the masking strategy for motion sequence reconstruction, HTM eliminates the need to train an encoder for representation extraction. Instead, it consists of two components with distinct purposes:
- The **HTM Transformer (Loss Predictor)**: Predicts reconstruction difficulty for each token in an unmasked motion sequence, guiding the masking strategy.
- The **Masked Transformer (Reconstructor)**: Reconstructs the masked tokens, concentrating on the most difficult segments identified by the loss predictor.

Unlike HPM’s emphasis on representation learning, HTM focuses entirely on improving reconstruction quality by selectively masking complex dynamics in motion data.

2) **Input Consistency in the Student-Teacher Model**

In HPM, the loss predictors in the teacher and student models operate on different inputs, resulting in input inconsistency:
- During training, the student model’s loss predictor learns from the encoded representation of masked images.
- The teacher model, however, uses the encoded representation of unmasked images to determine which patches to mask.

This can lead to suboptimal masking guidance, as the teacher's decisions are based on unmasked inputs that differ from the student's context.

However, HTM eliminates this issue by using the same unmasked token sequences for both teacher and student models of the HTM transformer, ensuring consistent masking guidance throughout the process. By maintaining consistent inputs, HTM simplifies its design and enables a more robust and reliable masking strategy.


**HGM³: Novel Contributions Beyond Existing Hierarchical Approaches**

HGM³ builds on hierarchical semantic graph reasoning but introduces key innovations that improve text-to-motion generation:

1) **Unified Weight Sharing Across Hierarchical Levels**

HGM³ employs a single model with shared weights across all hierarchical levels (motion, action, specifics):
- This significantly reduces the number of parameters, making the model faster and more memory-efficient.
- Weight sharing ensures that the model learns consistent, generalizable features across levels, enhancing coherence in the generated motions. Without shared weights, models often develop disjointed representations, leading to inconsistencies.

2) **Single Latent Space for Hierarchical Conditioning**

While GraphMotion (Jin et al., 2024) maintains separate latent spaces for different semantic levels, HGM³ unifies these into a single latent space. This simplifies the conditioning process, reducing computational overhead while preserving fine-grained semantic representation.

We believe these tailored innovations not only demonstrate originality but also significantly contribute to the field of motion generation.

---

> ### Comment · Reviewer_z3vH · 2024-11-26
>
> (1) I read the HPM (Wang et al., 2023a) paper, and I don't understand why you say HPM is designed to "learn novel representation" and "mask-invariant features." As far as I know, the goal of HPM is consistent with that of your HTM, which is to learn challenging patches (tokens) to improve the model's reconstruction ability.
> (2) When you say, "HTM eliminates the need to train an encoder for representation extraction," may I ask, what does the VQVAE in the first stage count as?

---

> > ### Comment · Reviewer_RKiq · 2024-11-27
> >
> > I am also interested in the answers to the questions raised here by Reviewer z3vH.

---

> > ### Author Response · Authors · 2024-11-27
> > **Response to Reviewer z3vH**
> >
> > >**Q-1)** On the Goal of HPM and "Mask-Invariant Features"
> >
> > **A)** Thank you for your review and for engaging with our response. To answer your question, we would like to distinguish between (1) the training objective of the model pre-trained in HPM and (2) the purpose of pre-training the model for downstream tasks.
> >
> > HPM is a self-supervised learning framework. In such frameworks, the goal of pre-training is to extract generalized representations that capture meaningful patterns in the input data. HPM achieves this by training the model to predict images in the masked patches, with the training objective being “to identify challenging patches to improve the model’s reconstruction ability.” However, the primary purpose of pre-training in HPM is **not to enhance reconstruction.** Instead, the focus is on enabling the encoder to learn robust representations as a byproduct of this process. For this reason, HPM explicitly trains an encoder as part of its framework.
> >
> > In practice, when the model encounters differently masked versions of the same image, it must consistently reconstruct the image regardless of the presence or location of masks. This requires the encoder to identify crucial patterns that are invariant to mask placement. This is what we mean by “mask-invariant features.” Once pre-trained, the encoder can then be directly used to extract generalized representations for downstream tasks.
> >
> > Our proposed HTM, in contrast to the HPM, is not aimed at training an encoder to extract generalized features for downstream tasks. Our focus lies on text-to-motion generation, where the model generates motion sequences by reconstructing masked tokens. Here, improving the ability to reconstruct masked tokens is not only the training objective but also the fundamental goal of the task—enhancing motion generation. As a result, unlike HPM, HTM does not involve training an encoder for representation learning.
> >
> > If the reviewer thinks that this is too strong terminology, we will tone it down as simply stating it as “robust features to masking”.
> >
> > >**Q-2)** On the Role of VQVAE in HTM
> >
> > **A)** Regarding the role of VQ-VAE in our framework, it is not trained as part of HTM but serves as a preprocessing step to quantize motion data into discrete tokens. Unlike the encoder in HPM, which is the primary learning target, the VQ-VAE encoder in our framework is fixed after its initial pre-training. HTM operates independently of this encoder, focusing entirely on improving the masked transformer’s reconstruction ability for motion sequences. Therefore, while VQ-VAE plays a role in tokenizing motion data, it is not a part of the encoder-driven representation learning process that is central to HPM.

---

### Comment · Reviewer_RKiq · 2024-11-20
**Some supplementary video files cannot be opened**

Dear authors,

Thank you for adding new supplementary video files.

Some of them cannot be opened.

Can you kindly re-upload them?

---

> ### Author Response · Authors · 2024-11-20
> **Response to Reviewer RKiq**
>
> We appreciate you bringing this issue to our attention and sincerely apologize for the inconvenience caused by the inaccessible video files. We have now re-uploaded the supplementary videos and verified that they can be opened successfully. Please let us know if you encounter any further issues.
>
> Thank you for your understanding and for taking the time to review our work.

---

### Meta-Review · Area_Chair_hrQv · 2024-12-19

**Metareview:**

This is a borderline paper. It exhibits strengths in writing, methodology, and experimental performance. However, there are still significant concerns regarding its relatively modest technical contributions and innovations, which is pointed out by all reviewers. The proposed method builds upon several existing techniques—for instance, hard token mining extends hard patch mining (Wang et al., 2023a), and the hierarchical graph representation is an extension of GraphMotion (Jin et al., 2024). Although the authors have expanded these modules and implemented detailed modifications to better address the problems presented, which is explained in their rebuttal, the overall technical contributions still appear marginal.

The Area Chair shares the reviewers' concerns about the paper's novelty. Nevertheless, considering its strengths in writing, methodology (its well-designed methodology that effectively extends and integrates existing methods), and comprehensive experiments, this paper falls within the borderline acceptance range but does not merit a clear acceptance.

**Additional Comments On Reviewer Discussion:**

The authors' rebuttal addressed many of the initial concerns and all reviewers reached marginal above the acceptance threshold rating. Yet, the issue of limited technique contributions is not fully addressed.

---

### Decision · Program_Chairs · 2025-01-22

Accept (Poster)